# Hidden in Plain Sight: Evaluating Abstract Shape Recognition in Vision-Language Models

**Arshia Hemmat**
University of Oxford

**Adam Davies**\*
University of Illinois Urbana-Champaign

**Tom A. Lamb**\*
University of Oxford

**Jianhao Yuan**\*
University of Oxford

**Philip Torr**
University of Oxford

**Ashkan Khakzar**
University of Oxford

**Francesco Pinto**
University of Oxford

## Abstract

Despite the importance of shape perception in human vision, early neural image classifiers relied less on shape information for object recognition than other (often spurious) features. While recent research suggests that current large Vision-Language Models (VLMs) exhibit more reliance on shape, we find them to still be seriously limited in this regard. To quantify such limitations, we introduce `IllusionBench`, a dataset that challenges current cutting-edge VLMs to decipher shape information when the shape is represented by an arrangement of visual elements in a scene. Our extensive evaluations reveal that, while these shapes are easily detectable by human annotators, current VLMs struggle to recognize them, indicating important avenues for future work in developing more robust visual perception systems. The full dataset and codebase are available at: `https://arshiahemmat.github.io/illusionbench/`

## 1 Introduction

Deep neural networks have accomplished remarkable breakthroughs in visual recognition over the past decade [Krizhevsky et al., 2012, He et al., 2016, Dosovitskiy et al., 2020, Radford et al., 2021, Gemini Team et al., 2023]; but these models have also shown longstanding, fundamental limitations – for instance, the performance of these models degrades when faced with common corruptions and perturbations [Hendrycks and Dietterich, 2019], or natural out-of-distribution data [Hendrycks et al., 2021]. How can we facilitate more robust neural vision models? A natural place to begin is by considering the source of robustness in human vision. Human object recognition is largely based on shape perception [Landau et al., 1988, Biederman and Ju, 1988, Xu et al., 2004, Baker and Kellman, 2018], which is essential to the robustness of human vision due to the invariance of shape to common transformations such as translation, rotation, scaling, and changes in illumination, color, and texture [Kendall, 1984, Hummel, 2001, Ommer, 2013, Dryden and Mardia, 2016]. As such, substantial work in computer vision has focused on improving and evaluating shape perception (e.g., Ritter et al., 2017, Geirhos et al., 2019, Islam et al., 2021, Geirhos et al., 2021, Gavrikov et al., 2024, *inter alia*), finding that early deep vision models relied much more on texture than shape in image classification [Geirhos et al., 2019, Islam et al., 2021, Pinto et al., 2022a, Benarous et al., 2023, Subramanian et al., 2023], which is believed to contribute to their lack of robustness [Geirhos et al., 2020, Gavrikov et al., 2024]. Later work observed that vision encoders trained with larger-scale data weakly supervised by language (e.g., CLIP; Radford et al., 2021) show improvements in shape recognition [Geirhos et al., 2021, Gavrikov et al., 2024].

While clear indicators of progress in visual perception of neural vision models, it is important to note that all of the above studies on shape recognition in vision models have relied on two standard datasets, Cue Conflict and Stylized-ImageNet [Geirhos et al., 2019], which presents several

---

\*These authors contributed equally to this work

38th Conference on Neural Information Processing Systems (NeurIPS 2024) Track on Datasets and Benchmarks.

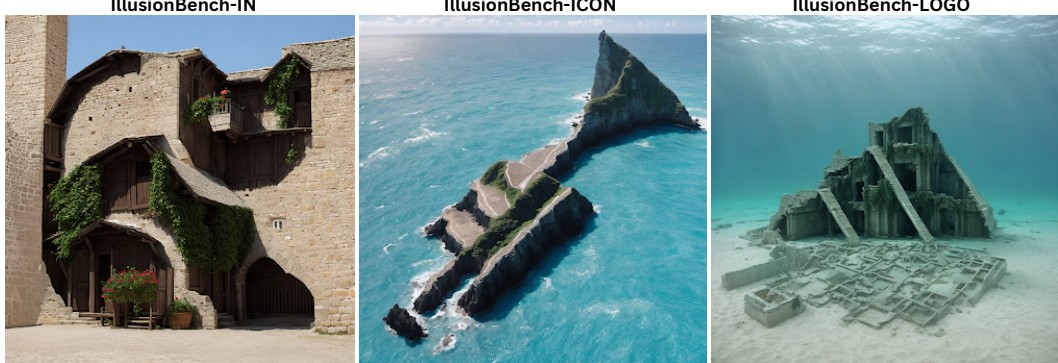

| IllusionBench-IN | IllusionBench-ICON | IllusionBench-LOGO |

Figure 1: **Can vision-language models recognize these shapes?** `IllusionBench` dataset contains images in which scene elements are arranged to represent abstract shapes.

concerns – for instance, these datasets do not include coherent, naturalistic visual scenes; they are built using legacy style transfer techniques that damage shape information and prevent the reproduction of fine-grained textures; and each image includes only a single object class represented as an abstract shape using perceptually uniform textures (see Section 2 for a more detailed critique). To address these limitations, we introduce `IllusionBench`,[1] which represents shape information by an arrangement of visual elements existing in coherent, naturalistic scenes (see Figure 1). We evaluate vision-language models (VLMs) using `IllusionBench` in three scenarios: (1) measuring **zero-shot** performance of generative VLMs (e.g., LLava [Liu et al., 2024b], GPT-4o [OpenAI, 2023], and Gemini [Gemini Team et al., 2023]); (2) measuring **few-shot** performance of VLMs using in-context learning (e.g., [Zhao et al., 2023]); and (3) **fine-tuning** contrastive VLMs (e.g., CLIP [Radford et al., 2021]) to recognize abstract shapes and testing their ability to generalize to unseen scenes. We find that, while human annotators can easily identify these shapes, VLMs struggle to identify shapes and instead focus on the scene components, failing to exhibit the abstract shape recognition capabilities that are essential for enabling humanlike visual robustness.

## 2 Background and Related Work

**Shape perception and visual recognition** Shape information is widely considered to be the most important cue leveraged by the human visual system for object recognition [Landau et al., 1988, Biederman and Ju, 1988, Xu et al., 2004, Elder and Velisavljević, 2009, Baker and Kellman, 2018]. Our ability to perceive shapes is crucial in enabling the robustness of human visual perception [Hummel, 2001, Ommer, 2013], as shape is invariant to key transformations such as translation, rotation, scaling, and changes in illumination, color, and texture [Ommer, 2013, Kendall, 1984, Dryden and Mardia, 2016]. Thus, many works have investigated the extent to which neural object classifiers rely on shape for visual recognition tasks, finding that early supervised deep neural networks rely more on texture cues rather than shape [Geirhos et al., 2019, Islam et al., 2021, Benarous et al., 2023, Pinto et al., 2022a, Subramanian et al., 2023]. More recently, Gavrikov et al. [2024] showed that multimodal vision-language models can be prompted to rely more on shape in visual recognition. Each of these works evaluates shape perception on the basis of the Cue Conflict (CC) or Stylized-ImageNet (SIN) benchmarks [Geirhos et al., 2019]; but despite their longstanding utility, we observe several key limitations with these benchmarks:

1. **Lack of coherent, naturalistic, and complex visual scenes:** Images contain only the shape of a single class mixed with a single texture applied uniformly to the entire image.
2. **Missing shape information:** Key shape information is often lost, yielding "a substantial fraction" of images that are unrecognizable to human annotators [Geirhos et al., 2019]. The contrast in textures between the object and the background of any given image is usually lost, yielding perceptually uniform images [Chen et al., 2021, Wang et al., 2023].

---

[1]We use "Illusion" in the name of our benchmark because images in our dataset can be understood as instances of pareidolia, an illusion caused by the tendency of the human visual system to identify familiar shapes in complex scenes. Our dataset should not be confused with HallusionBench [Guan et al., 2023], which instead serves as a diagnostic tool to distinguish between VLM reasoning error modes, such as those caused by the language component versus visual component of VLMs.

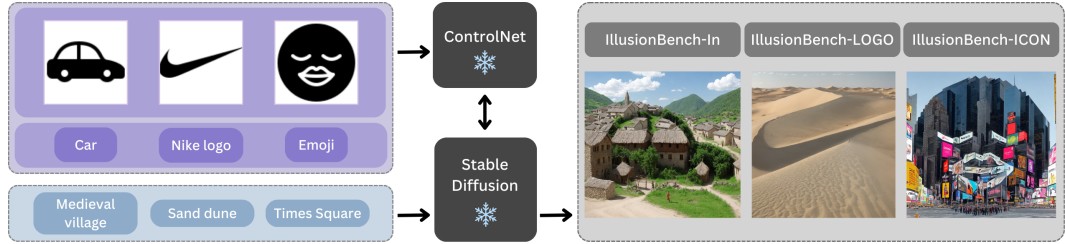

Figure 2: **Dataset generation.** For each of the 3 datasets in `IllusionBench`, we show an example image from the dataset alongside an example scene prompt and an example shape conditioning image used to generate it. A shape image $x_i$ (with the class name $c_i$) and a scene description $s_j$ are combined to generate the `IllusionBench` image $x_{ij}$.

3. **Low-quality style transfer:** The style transfer methods in these datasets [Gatys et al., 2016, Huang and Belongie, 2017] are known to confuse shape and texture information [Wang et al., 2023] and often fail to capture fine-grained textures [Wang et al., 2021].

To address these limitations, we introduce `IllusionBench`, which leverages state-of-the-art generative models to create images representing shape information with a complex arrangement of elements in detailed visual scenes comprised of various textures and objects.

**Evaluating visual capabilities of VLMs** Vision-language models (VLMs) have exceeded conventional benchmarks, often even exhibiting capabilities that they are not explicitly trained for [Bubeck et al., 2023] and underscoring the need for new forms of evaluation [Zhang et al., 2024]. Traditional image recognition benchmarks are not designed to characterize such capabilities, indicating the need for innovative evaluations. For instance, Bitton-Guetta et al. [2023] studies commonsense visual reasoning by testing whether models perceive peculiar content in visual scenes; Fu et al. [2024] evaluates VLMs on recognizing the count of objects, relative positions of objects, OCR, and commonsense visual reasoning; and Tong et al. [2024] proposes visual tasks requiring fine-grained understanding of object orientation, perspective, and the states of objects in the image. Finally, Zhou et al. [2023], Li et al. [2023d] focus on limitations specific to generative VLMs, such as visual hallucination.

## 3 Benchmark Description

### 3.1 Generative Process and Notation

Consider the set $\mathcal{C} = \{(x_i, c_i)\}_{i=1}^{|\mathcal{C}|}$ of binary shape conditioning images $x_i$ representing the shapes of corresponding object class $c_i$, and $\mathcal{T} = \{(s_j)\}_{j=1}^{|\mathcal{T}|}$ is the set of prompts where each $s_j$ describes a different scene (e.g., `Ocean` or `Medieval Village`). To synthesize our dataset, we use ControlNet [Zhang et al., 2023a], a module that is trained to control the generative process of text-to-image diffusion models (such as Stable Diffusion; Rombach et al., 2022) by conditioning on inputs specifying spatial information to guide the generative process, such as our shape conditioning images $x_i$ (refer to Figure 2 for an overview). The pipeline (Figure 2) transforms the tuple $(x_i, s_j)$ into an image $x_{ij}$ representing the considered shape $x_i$ of class $c_i$ embedded in a scene of type $s_j$.[2] We therefore obtain our datasets by creating a tuple $(x_{ij}, c_i, s_j)$ for each combination of conditioning images and prompts. We then consider three predictive tasks a VLM $f$ should perform (where $p_C, p_S$, and $p_{C,S}$ represent prompts querying for $c_i, s_j$, or both, respectively):

1. $\tau_C$: predict the shape $c_i = f(x_{ij}, p_C)$.
2. $\tau_S$: predict the scene $s_j = f(x_{ij}, p_S)$.
3. $\tau_{C,S}$, predicting both the shape and the scene $(c_i, s_j) = f(x_{ij}, p_{C,S})$.

### 3.2 Dataset Details

As exemplified in Figure 2, the `IllusionBench` benchmark contains three different constituent datasets: `IllusionBench-IN`, `IllusionBench-LOGO`, and `IllusionBench-ICON`. The number of

---

[2]The generation is conditioned on additional hyperparameters that allow us to obtain shapes that can be recognized at varying levels of abstraction. See Appendix B.2 for further details.

samples, classes, conditioning images, and domains for each dataset are provided in Table 1 (with more detailed metadata available in Appendix B).

Table 1: Size of each dataset in `IllusionBench`.

| Dataset Name | # Samples | # Classes | # Conditioning Images | # Scenes |
|---|---|---|---|---|
| IllusionBench-IN | 6864 | 16 | 48 | 11 |
| IllusionBench-LOGO | 5577 | 21 | 39 | 11 |
| IllusionBench-ICON | 20064 | 6 | 456 | 11 |

`IllusionBench-IN` We build upon the 16 classes from the most popular shape perception benchmark, Stylized-ImageNet (SIN) [Geirhos et al., 2019]. However, since we are interested in how well models can find shapes within a scene, we need clear and distinct shapes that can be identified unambiguously. To address this, we replace 4 of the 16 SIN classes with similar categories (near co-hyponyms) with more distinct shapes. We collect 3 conditioning images for each class.

`IllusionBench-LOGO` Another category of shapes that are specifically designed to be visually distinct and easily recognizable are logos, which provide an interesting contrast to the shapes in `IllusionBench-IN`, as recognizing them requires world knowledge specific to the category of product brands (rather than culturally-nonspecific real-world object classes).[3] Thus, we expand our dataset to this domain by collecting 39 different logo conditioning images across 21 brands.

`IllusionBench-ICON` Finally, we develop a third dataset to test whether VLMs can be trained to recognize cross-modal abstractions over perceptually distinct shapes representing semantically related concepts (e.g., where images representing shapes of owls or turtles are both recognized as instances of the "animal" class, despite having very different shapes). We create a coarse-grained dataset of 6 (informal) hypernym categories across 456 emojis as shape conditioning images.

**Validating Dataset Quality** Although ground truth labels for object classes and scene types are available, image generators may sometimes produce low-quality or high-difficulty images whose object shape is not human-recognizable. To minimize the proportion of such images, we begin by restricting the hyperparameters that control the influence of the conditioning image to ranges that we qualitatively found to produce clearly distinguishable shapes (see Appendix B.2). To validate that the shapes in the resulting images are indeed human-recognizable, we recruited 60 participants (information is anonymized) to manually annotate randomly sampled subsets of `IllusionBench-IN`, `IllusionBench-LOGO`, `IllusionBench-ICON`, obtaining an average annotator accuracy of 95.6%, 97.17% and 96.8%, respectively, indicating that humans are indeed able to recognize the shapes in the vast majority of the generated images.[4] (See Appendix B.1 for further details.)

### 3.3 Evaluation

Given image $x_{ij}$, we prompt VLM $f$ with both $x_{ij}$ and prompts $p_k$ corresponding to the shape, scene, and both the shape and scene (i.e., where $p_k$ is variously $p_C, p_S$, or $p_{C,S}$, respectively), yielding responses $r_k = f(x_{ij}, p_k)$ for each prompt $p_k$. For each $x_{ij}$, we evaluate *shape recall* on the basis of whether the term $c_i$ appears in the response $r_C$ or $r_{C,S}$ (yielding 1 if so, or 0 if not), and evaluate *scene recall* by whether $s_j$ appears in $r_T$ or $r_{C,S}$ (similarly yielding 1 or 0), and report the shape and scene recall for each dataset as the sum of the recall figures across all $x_{ij}$ instances divided by the size of each dataset. In contrast to prior related works (e.g., Geirhos et al. 2019, 2021, Gavrikov et al. 2024), our proposed metrics are designed such that shape recognition performance is not in competition with the ability to recognise other visual elements (e.g., textures or scene elements), as – unlike traditional classifiers, which must select only one among a pre-defined set of discrete classes

---

[3]Given that this task requires both world knowledge of product brands *and* abstract shape recognition capabilities, and considering that our goal with `IllusionBench` is only to evaluate the latter, we normalize scores by averaging results for each VLM exclusively on samples obtained from raw shapes that the VLM can recognise in a zero-shot setting, meaning that models are not penalized for lacking world knowledge of specific brands. See Appendices C.5 and D.7 for non-normalized results by class.

[4]Note that human annotator accuracies are only intended to *validate the quality of the generated dataset* and *confirm that the resulting abstract shapes are indeed human-perceptible*. They are *not* intended for direct comparison with VLM performance, as there are a few fundamental differences in how annotators and VLMs are tested. For instance, where VLMs do not know the purpose or structure of the task beyond what is included in the prompt, annotators are shown onboarding materials describing the task, including several pre-annotated examples.

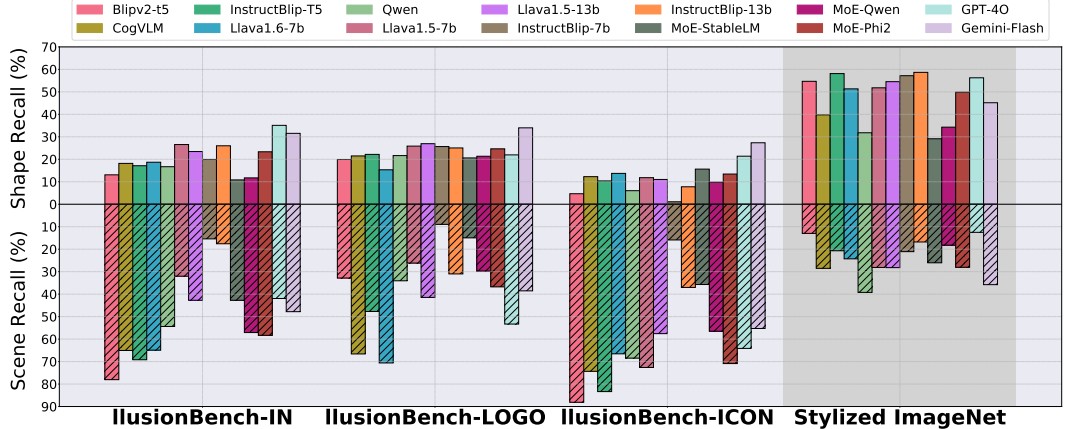

Figure 3: **Zero-Shot Results**. Average shape and scene recall of VLMs across each `IllusionBench` dataset, compared with Stylized-ImageNet [Geirhos et al., 2019] (rightmost, shaded).

– generative VLMs can respond with detailed descriptions of images including information about shape, scene, or other visual elements at the same time (or given different prompts).

### 3.4 Experimental Overview

In the following sections, we evaluate the shape perception capabilities of modern VLMs on `IllusionBench` under the following paradigms:

- **Zero-Shot Recognition**: Given that an instruction-tuned VLM can recognise a shape $x_i$, can it identify the same shape when it emerges from the combination of visual elements in $x_{ij}$ without any explicit examples or specialized fine-tuning? (Section 4)
- **Few-Shot Learning**: Given that a multi-modal in-context learner can recognise a shape $x_i$ zero-shot, can it leverage few examples to learn to identify it in $x_{ij}$? (Section 5)
- **Domain Generalization**: Given training samples $\{x_{ij}\}$ representing a shape $x_i$ in certain types of scenes, can models learn to recognise the same shape in other, unseen scene types? (Section 6)

## 4    Can Instruction-Tuned VLMs Recognize Shapes Zero-Shot?

**Experimental Design**   In this experiment, we prompt VLMs zero-shot to identify the abstract shape represented in a visual scene among a closed set of object classes. We begin by testing whether models can correctly classify the shape conditioning images (binary shape images), and generate images for `IllusionBench` exclusively using these condition shapes. We then prompt models with respect to the shape and scene in each generated image, and measure the corresponding recall metrics as described in Section 3.3. (See Appendix C for additional details regarding the experimental design, prompts, and models used in this experiment.)

**Models**   We consider the following VLMs for evaluation: `GPT-4o` [OpenAI, 2023], `Gemini-Flash` [Gemini Team et al., 2023], `LLaVA1.5/6-7/13b` [Liu et al., 2024c], `CogVLM` [Wang et al., 2024], `BLIPv2-t5` [Li et al., 2023c], `InstructBLIP-7/13b` [Dai et al., 2024], `Qwen-VL-Chat` [Bai et al., 2023], and `MoE-StableLM/Qwen/Phi2` [Lin et al., 2024].

**Results**   Our main findings in this experiment (visualized in Figure 3) are as follows:

- For each of our datasets, shape recall is quite low, with most models ranging between 10-30% (in contrast to the previous dataset, Stylized-ImageNet [Geirhos et al., 2019], where all fourteen models exceed 30%).
- For nearly all models and datasets, models exhibit superior scene recall relative to shape recall. This indicates that the recognition capacity of current VLMs is still biased towards scene/texture features, similar to earlier work studying CNN classifiers (see Section 2).
- `GPT-4o` and `GEMINI` show superior shape recall to all other models in 3/3 and 2/3 of our datasets, respectively, demonstrating a shape-recognition gap between the best available open- and closed-source VLMs.

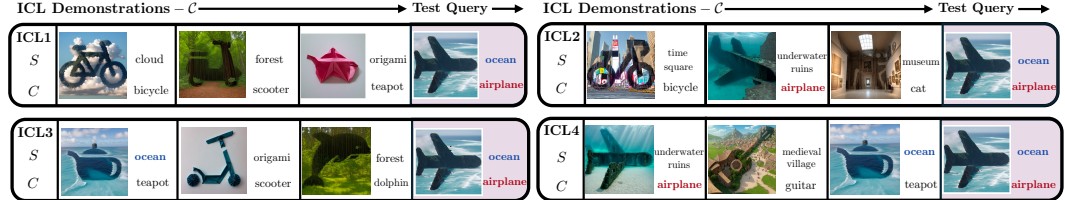

Figure 4: **ICL Learning Tasks**. Figure depicting the four ICL learning tasks, $ICL1, ICL2, ICL3$ and $ICL4$, defined by constraints on demonstration example selection as introduced Section 5.

- Mixture of Experts (MoE) (like `MoE-StableLM`, `MoE-Qwen`, and `MoE-Phi2`), which are generally employed to improve models' performance, exhibit neither superior shape nor scene recall with respect to individual models or closed-source models.
- Among all open source models, `LLava` attains the strongest shape recall performance across all our datasets. In contrast, `Blipv2` attains the highest scene recall (except for the `IllusionBench-LOGO` split).

See Appendix C for more fine-grained results and analysis.

## 5 Can In-Context Learners Learn to Identify Abstract Shapes?

Given zero-shot prompting exhibits poor performance at detecting abstract shapes and shows VLMs mostly focus on background stimuli, a natural question is whether it is possible to teach models to recognise known shapes with a few samples by leveraging their In-Context Learning (ICL) or few-shot capabilities.[5]

**Experimental Design** We restrict our experiments to samples generated from conditioning images $x_i$ that models can correctly classify in a zero-shot fashion (see Appendix D.2). Let us focus on the predictive task $\tau_C$ (as analogous formulations of ICL apply for $\tau_S$ and $\tau_{C,S}$). Given that the model can correctly assign the class $c_i$ to the conditioning image $x_i$, we provide it with the context sequence $\{(x_{i_w,j_w}, c_{i_w})\}_{w=1}^{|W|}$, where $W$ is the context window plus a test image $x_{i_*,j_*}$, and prompt the model to predict the object's shape $c_{i_*}$.

Using specific constraints on context sampling relative to a test sample, we define four learning tasks corresponding to perceptual challenges:

- **ICL1**: *Given the context lacks any image depicting the scene or shape type of the test sample $x_{i,j}$, can the model recognize its shape $c_i$?*
- **ICL2**: *Given the context includes an image of the shape type but not the scene type of the test sample $x_{i,j}$, can the model recognize its shape $c_i$?*
- **ICL3**: *Given the context includes an image of the scene type but not the shape type of the test sample $x_{i,j}$, can the model recognize its shape $c_i$?*
- **ICL4**: *Given the context includes images of the scene type and shape type of the test sample $x_{i,j}$ (separately and exactly once), can the model recognize the test sample's shape $c_i$?*

Samples in the context are selected uniformly at random, excluding those that do not satisfy the constraints for a given test sample. Random selection serves as a simple baseline for ICL example selection, avoiding confounding factors like similarity bias or majority [Bertini Baldassini et al., 2024]. We perform $0, 1, 2, 4, 8$-shot on `IllusionBench-LOGO` and `IllusionBench-IN`, and $1, 2, 4, 5$-shot on `IllusionBench-ICON`. Further details of ICL experiments can be found in Appendix D.2. We additionally perform ablations to examine the sensitivity of our results to the prompt template used or to the order in which in-context examples are given to the model. These additional results can be found in Appendix D.9.

**Models.** We consider several state-of-the-art models that have been designed to support ICL: (1) `LLaVA-Next` [Liu et al., 2024b], (2) `Qwen-VL-Chat` [Bai et al., 2023], (3) `Otter-MPT` [Li et al., 2023a], (4) `IDEFICS-9B-Instruct` [Laurençon et al., 2024], and (5) `MMICL-T5-XXL`[Zhao et al., 2023]. (We describe each models, the prompts they are provided, and a detailed motivation for selecting these particular models in Appendix D.3.)

---

[5]See Appendix D.1 for a brief introduction to ICL.

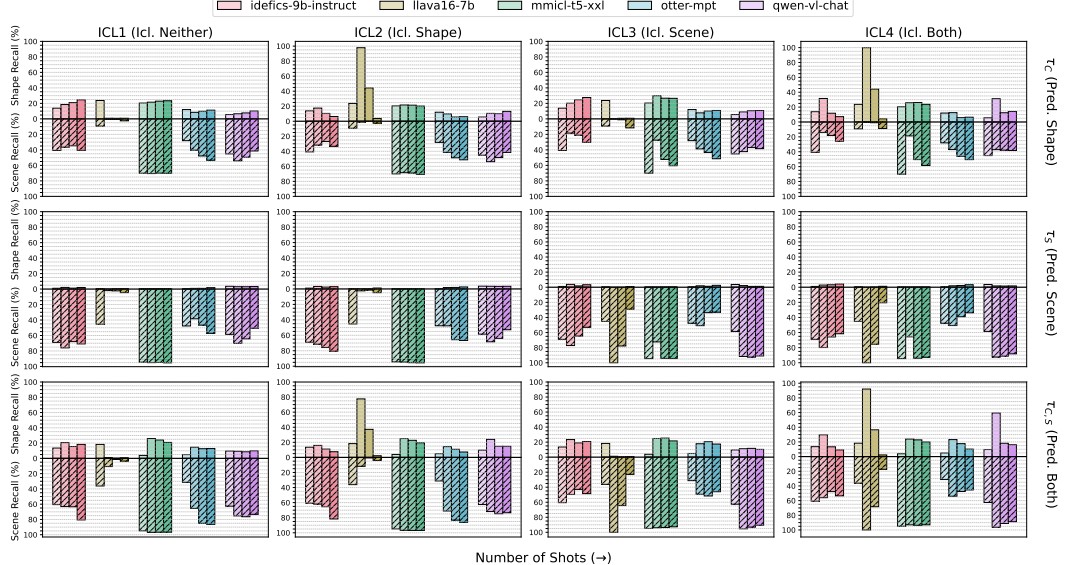

Figure 5: **ICL Results**. Few-shot (0,1,2 and 4-shot) shape and scene recall of VLMs averaged across the `IllusionBench-LOGO`, `IllusionBench-IN` and `IllusionBench-ICON` datasets, displayed for the different ICL learning tasks and the different prediction tasks.

**Results.** We summarise the average results across all three dataset splits for $0, 1, 2$ and $4$-shot ICL as show in Figure 5 following the recall metrics introduced in Section 3.3 (for results for individual datasets and for $5$-shot and $8$-shot performance on `IllusionBench-ICON` and `IllusionBench-IN/ IllusionBench-LOGO` respectively, see Appendix D.7.) [6] We report here the main trends in the data. Discussion of exceptions that do not follow the reported general trends can be found in Appendix D.6.

- *ICL does not mitigate tendency to predict scene over shape.* As shown in Figure 5, ICL has minimal effect in altering the models' tendency to predict the scene $s_j$, regardless of whether the prediction task is $\tau_C$ (predict shape), $\tau_S$ (predict scene), or $\tau_{C,S}$ (predict both).
- *On average,* MMICL-t5-XXL *exhibits the strongest scene and shape recall for the highest number of shots* (i.e., when majority voting biases decay; see [Bertini Baldassini et al., 2024]).
- *Increasing the number of shots has mixed effects on performance.* We observe in Figure 5 that the models often exhibit non-monotonic performance trends for both shape and scene recall across all prediction tasks and demonstration selection constraints. In general, this indicates that the models struggle in general to adapt to tasks $\tau_C$, $\tau_S$, and $\tau_{C,S}$, even with increasing demonstration examples. These results are in line with previous findings that complex ICL tasks remain challenging for current visual language models (VLMs) [Zong et al., 2024].
- *Context selection strategy effects prediction tasks differently.*
    - $\tau_C$ *(shape prediction):* As shown in the top row of Figure 5 for task $\tau_C$ (shape prediction), including the shape in the context (ICL2 and ICL4) either maintains or reduces performance for most models such as MMICL and IDEFICS. This suggests that most models struggle to identify and disentangle shape from the scene through ICL.
    - $\tau_S$ *(scene prediction):* The second row of Figure 5 shows the mixed effect of including the scene within the context (ICL3 and ICL4) compared to not including it (ICL1 and ICL2). Models such as LLAVA, OTTER show a reduction in scene recall and when including the scene in the context. MMICL maintains comparable performance, whereas LLaVA and QWEN show improved performance.
    - $\tau_{C,S}$ *(predicting both shape and scene):* The final row of Figure 5 typically shows trends similar to $\tau_C$ and $\tau_S$ – e.g., scene recall and shape recall for MMICL (whose zero-shot shape recall is lower on this task than in $\tau_C$), IDEFICS, and LLaVA are comparable with respect to those in $\tau_{C,S}$ and $\tau_S$ (respectively).

---

[6]In Figure 5, we observe that LLaVA shows often close to zero recall on either shape of scene prediction. We explore a few possible reasons for these results in Appendix D.8.

Overall, we observe that ICL does not substantially aid models in learning to detect abstract shapes within scenes or to help reduce scene prediction bias. The non-uniformity of relative results between models further highlights the immaturity of ICL for multi-modal models, particularly for complex tasks like abstract shape recognition.

## 6 Can VLMs Learn Invariant Representations Across Domains?

A compelling application of `IllusionBench-IN` lies in Domain Generalisation (DG) [Gulrajani and Lopez-Paz, 2020]. A visual domain is a set of samples with shared characteristics that influence the appearance of objects (e.g., shared style, such as cartoons, paintings, or photos; shared lighting conditions, such as photos taken at similar times of day with similar weather conditions; etc.). In DG, the goal is for models to learn domain-invariant representations – i.e., generalisable features that are predictive of task labels across any domain – by training across multiple "source" domains and testing how well models generalise to unseen test domains. (See Appendix E.1 for a more detailed introduction to DG.)

**Experimental Design.** We consider all images generated using the same scene prompt $s_j$ as coming from the same domain $\mathcal{D}^j$. As shown in Figure 6, we partition the `IllusionBench-IN` dataset split into train domains $s_j \in \{$Cloud, Forest, Ocean, Origami, Sand Dune$\}$ and test domains $s_j \in \{$Bazaar Market, City, Medieval Village, Museum, Times Square, Underwater$\}$. (Conditioning images $x_i$ used to generate the training domains are not contained in the test domains.) We then consider a contrastive language-vision encoder (CLIP [Radford et al., 2021]) and prompt CLIP in order to identify the class $c_{i_*}$ of a test sample $x_{i_*}$ among all possible shape classes[7]. Throughout the experiment, we use "A photo of {class_name}" as the prompt template. (See Appendix E.2 for further experimental details.)

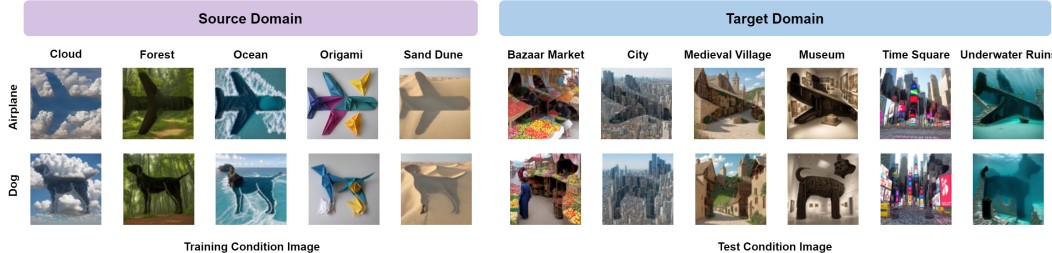

Figure 6: `IllusionBench-IN` **for Domain Generalisation.** We split the dataset into five source domains for training and six target domains for testing. The condition images for generated data samples are only shared among source and target domains, respectively, without overlapping.

**Methods Considered.** We compare various domain generalisation methods including ERM, MixUp [Yan et al., 2020], RegMixUp [Pinto et al., 2022b], GroupDRO [Sagawa et al., 2019], and VREx [Krueger et al., 2021], using both linear probing and full-parameter finetuning. Besides linear probing, we also consider DPLCLIP [Zhang et al., 2023b], a prompt optimization approach specifically designed for CLIP domain generalisation.

**Results.** We summarise our findings (reported in Figure 7) as follows:

- *CLIP cannot recognise shapes well in a zero-shot setting.* The CLIP model attains on average extremely low performance in zero-shot settings, with the exception of the Museum domain. This can be attributed to the fact that certain samples within this domain do not simply assemble $c_i$ from visual cues of other objects, but incorporate it as a sculpture.
- *CLIP embeddings only partially capture shape information.* Applying prompt learning for domain generalisation via DPLCLIP is not particularly effective with an average test accuracy of 13.62%, and ERM results are more effective in improving over the zero-shot performance with accuracy 22.36%, outperforming all other probing techniques. However, the relatively low absolute values of accuracy indicate the embedding space does not render the test samples linearly separable based on shape criteria.

---

[7]Since the zero-shot performance is particularly low, we do not confuse the model further asking it to distinguish the shape from the background type. This also allows us to make the comparison with probing and fine-tuning techniques that deliberately aim at extracting shape more fairly.

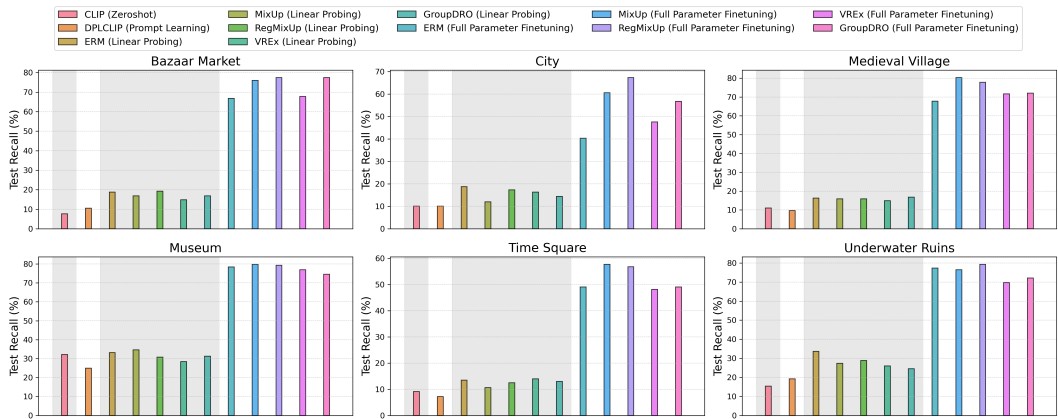

Figure 7: **Domain Generalisation.** CLIP performance on `IllusionBench-IN` for different fine-tuning approaches. Each sub-figure represents an unseen test domain. The categories of approaches, with alternating shading from left to right to indicate these different categories, are: zero-shot prediction, prompt learning, linear probing, and full parameter fine-tuning.

- *CLIP can learn features that allow to distinguish objects based on shape using multiple domains.* In full-parameter fine-tuning, it is possible to learn representations that are more oriented towards shape recognition – in all cases, a very large improvement is observed with respect to linear probing. The best performing methods are Mixup and RegMixup, which attain 71.79% and 73.00% on average accuracy, respectively.

## 7 Social Impact

The limited shape perception abilities of current vision systems, as highlighted in our work, could hypothetically be exploited by malicious users to, for instance, disseminate hateful or sensitive material online by bypassing inappropriate content visual filters that cannot recognize human-perceptible abstract shapes in scene elements (as enabled by the data-generation methodology we explore in this work). Conversely, improving perception ability could also aid censorship by moderators. In general, we anticipate that shape recognition capabilities on-par with generative techniques would empower platforms relative to users (e.g., for both content moderation and potential censorship), and shape recognition capabilities that are not able to recognize abstract shapes in outputs of leading generative techniques (as we observe in this work) empowers users relative to platforms, irrespective of whether content is legal or ethical.

## 8 Conclusion

We present `IllusionBench`, a collection of 3 datasets to evaluate shape recognition in vision-language models (VLMs) by representing abstract shapes as complex arrangements of visual scene elements. While human annotators identify these shapes with high accuracy, we find that state-of-the-art VLMs fail to identify the shapes in these scenes zero-shot, tending to focus on scene elements instead. We observe that in-context learning does not significantly improve models' ability to detect abstract shapes; but we do find that contrastive VLMs such as CLIP can be fine-tuned to recognize these shapes and generalize to new scene domains. In highlighting the limited shape perception abilities of current VLMs, we hope that `IllusionBench` will help guide future research in developing more robust computer vision systems. The contributions of each author are listed in Appendix A.

## Acknowledgements

This work is supported in part by the National Science Foundation and the Institute of Education Sciences, U.S. Department of Education, through Award #2229612 (National AI Institute for Inclusive Intelligent Technologies for Education). Any opinions, findings, and conclusions or recommendations expressed in this material are those of the author(s) and do not necessarily reflect the views of National Science Foundation or the U.S. Department of Education.

Ashkan Khakzar and Philip Torr are supported by UKRI grant: Turing AI Fellowship EP/W002981/1, and by the Royal Academy of Engineering (United Kingdom) under the Research Chair and Senior Research Fellowships scheme.

We would like to express our gratitude to Yawei Li for performing some preliminary experiments on a different topic (not included in this work) before we converged on the research topic explored in this work. We also extend our thanks to Ali Ma'manpush, Julia Hockenmaier, and Prashant Jayannavar for their invaluable assistance and advice regarding the human data annotation process.

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

# A  Author Contributions

- **Arshia Hemmat:**
  - *Dataset Generation:* Responsible for the development of the Illusion Generation Pipeline, meticulously designing and implementing the system to create high-quality illusions. Efforts included fine-tuning the hyperparameters to ensure the generated images met the desired standards of clarity and effectiveness. This process involved extensive experimentation and adjustment to achieve optimal results.
  - *Experiments and Method:* Conducted the zero-shot experiments for all the models, which involved setting up, running, and analyzing the results of these experiments.
  - *Data Annotation:* Managed data annotation involving over 100 individuals, collected and analyzed the annotation results to ensure data quality and relevance.
  - *Paper Writing:* Co-writer of the basics of the chapter Section 4.
- **Adam Davies:**
  - *Experiments and Method:* Co-supervised benchmark design, generation, evaluation, zero-shot experiments, and results analysis/visualization; implemented experimental prototypes for VLM generation and evaluation; assisted in model selection and deployment.
  - *Data Annotation:* Designed data annotation procedure; wrote onboarding materials for annotators; prototyped data annotation setup and assisted in deployment and analysis.
  - *Paper Writing:* Co-wrote, edited, and revised all sections of the paper; co-shaped central story (motivation, contribution, relationship with prior work in computer and human vision, results analysis, social impact); literature review (shape recognition in human vision, shape recognition benchmarks).
- **Tom A. Lamb:**
  - *Experiments:* Co-shaped design of ICL experiments; implemented and carried out all ICL experiments and co-led the analysis and presentation of ICL results.
  - *Paper Writing:* Led writing of Section 5. Additionally, contributed to the writing and presentation of Section 4.
- **Jianhao Yuan:**
  - *Experiments:* Carried out all domain generalization experiments Section 6. Additionally, carried out zeroshot experiment of GPT-4o and Gemini in Section 4.
  - *Paper Writing:* Led writing of Section 6.
- **Philip Torr:** Provided feedback and advice regarding the project direction and proposed approach; assisted in securing compute resources to carry out experiments.
- **Ashkan Khakzar:**
  - *Idea* Conceived the research problem and idea (the idea to evaluate VLMs on shapes represented by an arrangement of visual scene elements)
  - *Method* Identified the existing method to generate such images. Demonstrated proof of concept (that state-of-the-art VLMs cannot identify these shapes)
  - *Literature review* On shape recognition in computer vision and co-shaped the storyline
  - *Experiments* Project co-supervision (curating the dataset Section 3, and zero-shot experiments Section 4)
  - *Writing* Co-writing of abstract, introduction, related works, and conclusion.
- **Francesco Pinto:**
  - *Experiments and Method:* Led benchmark design, generation and evaluation; led design, results analysis and visualization of zero-shot, in-context learning and multi-domain generalization experiments.
  - *Data Annotation:* Implemented and tested data annotation procedure, assisted in preparing the onboarding materials for annotators; collected and analysed the results of the annotation.
  - *Paper Writing:* Co-shaped central story (motivation, contribution, relationship with prior work in computer vision, results analysis, impact); literature review (on shape recognition in computer vision); co-wrote the abstract and all sections of the paper but conclusions. Co-supervised full project.

# B  Dataset Documentation and Additional Information

Below, we include all information required for dataset submissions to the NeurIPS Datasets and Benchmarks Track:

**Dataset Documentation and Intended Uses**  The dataset documentation is provided at the Croissant and Huggingface URLs mentioned below. The dataset mainly evaluates foundational VLMs and their shape recognition abilities. The dataset can also learn invariant representations using domain generalisation techniques. Other uses may be possible.

**Dataset URL**  Our datasets are available for viewing and full download at the following permanent link: `https://huggingface.co/datasets/arshiahemmat/IllusionBench`. The "dataset viewer" allows one to select a specific split (i.e., `IllusionBench-IN`, `IllusionBench-LOGO`, or `IllusionBench-ICON`). All images are provided in the `.png` format. The HuggingFace Datasets repository service (where our dataset is hosted) automatically generates structured Web standard metadata for dataset discovery.

**Croissant Metadata URL**  Our Croissant metadata record is available at `https://huggingface.co/api/datasets/arshiahemmat/IllusionBench/croissant`.

**Author Statement**  The authors have collected the conditioning images and generated this dataset for research purposes. For this reason, the data usage is allowed under the fair use law and is not intended to yield any copyright infringement. There is no warranty of fitness for a particular purpose or noninfringement. The authors remain available to edit the dataset to comply with the law. In no event shall the authors or the NeurIPS conference be liable for any claim, damages, or other liability arising from, out of, or in connection with the usage or release of this dataset.

**Data License**  This work is openly licensed under CC BY-NC 4.0 (`https://creativecommons.org/licenses/by-nc/4.0/deed.en`).

**Long-Term Hosting, Licensing, and Maintenance Plan**  We have uploaded our dataset to Hugging-Face Datasets (link above). The Licensing information and Croissant metadata URL are available above and also available in the HuggingFace URL. Regarding Maintenance of the dataset on the HuggingFace servers please refer to the `https://huggingface.co/content-guidelines`.

**Reproducibility**  The code for generating the dataset and the experiments are publicly available in the following repository `https://github.com/arshiahemmat/IllusionBench`.

**Human Annotations**  We have provided screenshots of annotation forms which were distributed among participants in Appendix B.1.

**Attributions**  This work utilizes stock images to condition generators (as described in Section 3). `IllusionBench-ICON` conditioning images are taken from `icons8.com`, which makes them freely available provided they are attributed using a link (as we do here).

## B.1  Human Annotation Details

**Subsampling for Annotation**  Given the size of our dataset ( more than 32K samples) performing a complete annotation of it would be expensive. Furthermore, since the data is synthesized and we perfectly know the class of the shapes represented in each image, the purpose of the annotation is simply to verify that the generated images have shapes that are recognisable by humans. For this reason, we subsample the generated dataset by enforcing that, for each dataset (i.e., each of `IllusionBench-IN`, `IllusionBench-ICON`, and `IllusionBench-LOGO`), at least one conditioning image from each class and scene choice is annotated.

Furthermore, we observe that the difficulty in perceiving an object depends on the choice of the hyperparameters that control the diffusion process. For this reason, we additionally enforce that images are uniformly sampled from each hyperparameter setting so that annotators are exposed to images encompassing the full range of difficulty.

**Participants**  Our human evaluation involved 106 participants. The annotators were first instructed about the task and required to perform a simple test on 10 images, in order to make sure they understood the task to be performed. Annotators participated on a purely volunteer basis and were awarded with in-course credit. Participation was not mandatory for any student or course. No risks were identified for the annotation process.

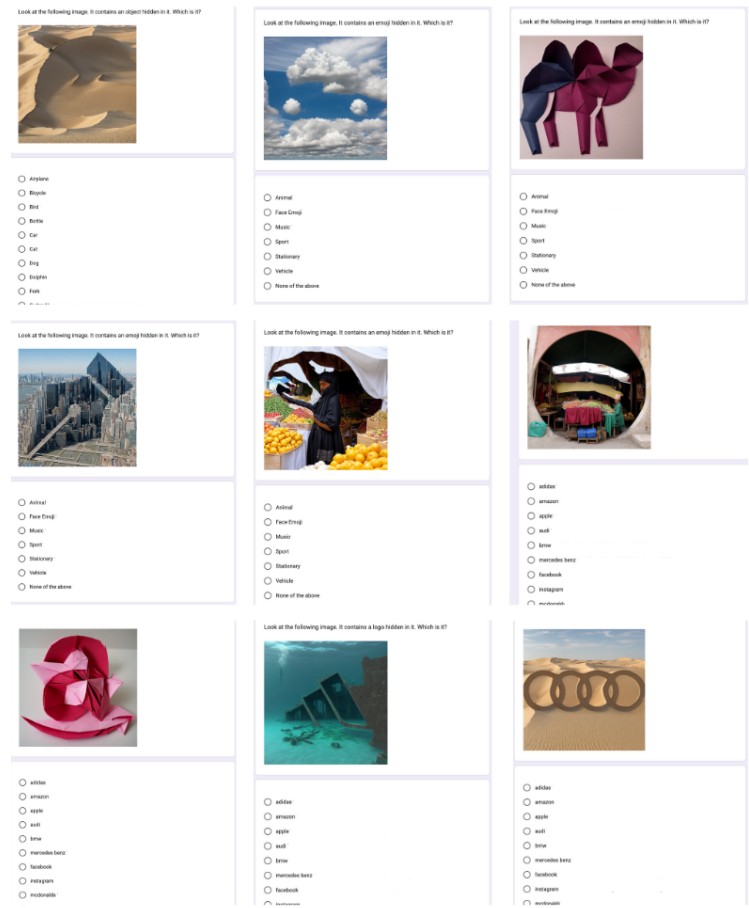

Figure 8: **Human Annotation** screenshots from GoogleForm through which the images are annotated by human annotators.

**Further annotation meta-data** The number of annotators for `IllusionBench-IN`, `IllusionBench-ICON`, and `IllusionBench-LOGO` is respectively of 35 each. We split the original data so that each sample is annotated twice. Each reviewer is assigned approximately 80 samples (since the splitting algorithm is randomised, they may receive slightly less or slightly more samples).

## B.2 Image Generation Hyperparameters

For data generation, we focused on the Illusion Diffusion generative models (demo available here), containing three major components:

- ControlNet [Zhang et al., 2023a], specifically: controlv1p sd15 qrcode monster
- Base Model, specifically: RealisticVision V5.1 noVAE, built using Stable Diffusion [Rombach et al., 2022]
- Stable Diffusion-guided VAE, specifically: sd-vae-ft-mse

We used the following generation hyperparameters:

- Prompts were simply a single word corresponding to the scene types (e.g., "city" or "museum")
- Guidance-scale was always set to default value **7.5**
- Illusion_strength, which can be used to modulate the strength of abstract shape patterns, was selected based on our anecdotal observations regarding an appropriate difficulty level for each dataset (see below) and validated using human data annotation (as described above)
- Sampler was always set to default value **Euler**

The Illusion_strength for the different datasets are as follows:

- {Illusion_strength} of the `IllusionBench-LOGO` and `IllusionBench-IN`: [0.75, 0.80, 0.85, 0.90, 1.05, 1.10, 1.15, 1.20, 1.25, 1.35, 1.40, 1.50, 1.60]

- {Illusion_strength} of the `IllusionBench-ICON`: [0.85, 1.05, 1.25, 1.40]

### B.3 Limitations

For future work, we will create more complex images and define more tasks in order to challenge models. We have also increased the size of our dataset so that we can train large models using our dataset. A current limitation is that we only hide a single shape in each image. Future work could extend this to incorporating several objects within the same background. Finally, we also plan to experiment with further tasks for compositional understanding and scene understanding of SOTA models. We leveraged prompt engineering to report the best possible performance of each model in the zero-shot case as described in Appendix C and Section 4, however, improvements may be possible. We describe several limitations of the methods explored in this work in Sections 4 and 5.

### B.4 Data Samples

To illustrate the quality of abstract shape recognition images created for this dataset, we randomly sample one image from several scene types in each dataset and display them in Figure 9.

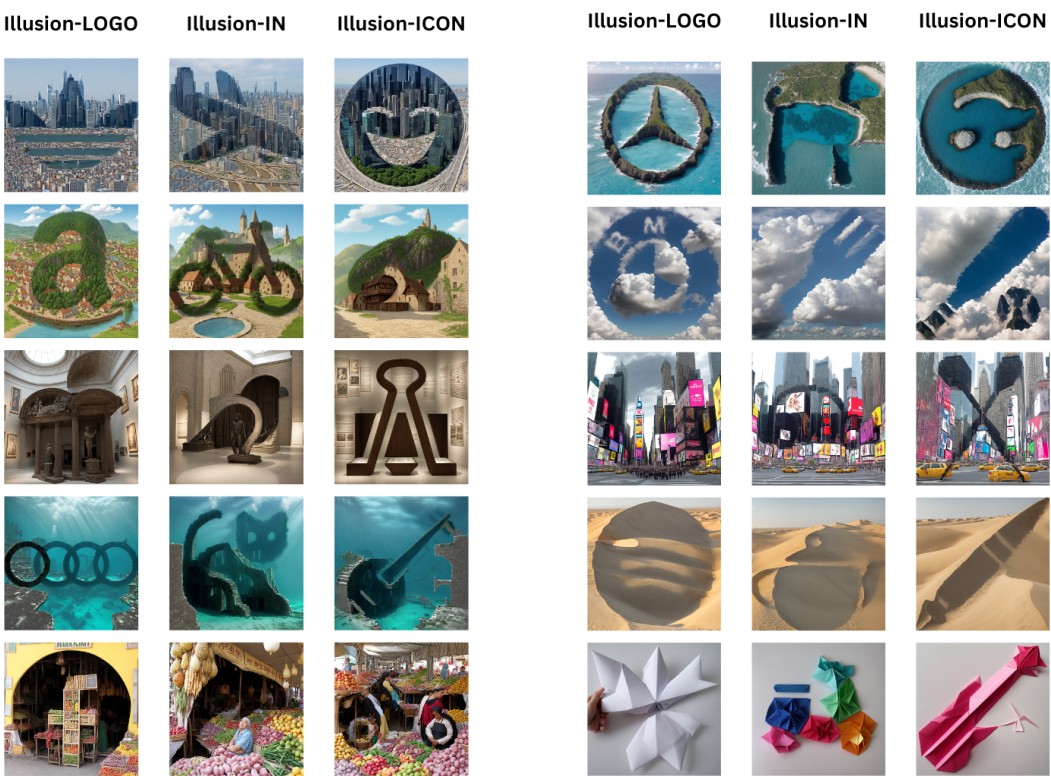

Figure 9: **Image Samples** from each dataset in our benchmark.

## C  Zero-Shot Experiments Details

### C.1  Zero-shot Experiments

We test our models zero-shot to evaluate their abstract shape recognition abilities. To leverage all capabilities of these models, we describe the conditions of our experiments in our prompt. The

models are then asked to choose the correct shape type among a closed set of options, which include both shapes and scene names.

Let us focus on the predictive task $\tau_C$. Analogous formulations hold for $\tau_S$ and $\tau_{C,S}$. Given that the model can correctly assign the class $y_{i_*}^C$ to the hidden shape in the scene $x_{i_*}^C$, we provide it with a set of options $\mathcal{O}$, which includes all the shapes and scene names considered in the dataset split. We then ask the model to predict the shape name from these options.

Define $\mathcal{O} = \{\text{shape}_1, \text{shape}_2, \ldots, \text{scene}_1, \text{scene}_2, \ldots\}$ as the set of possible options. The model's response is evaluated based on whether the correct shape name is present in its output.

## C.2 Models

In our zero-shot experiments, we evaluate each of the following large vision language models (VLMs):

- BlipV2-T5 [Li et al., 2023c], a VLM utilizing the T5 architecture [Raffel et al., 2020] for text encoding and a state-of-the-art vision encoder, designed for high-performance multimodal tasks.
- CogVLM [Wang et al., 2024], an advanced VLM leveraging a Vision Transformer (ViT) [Dosovitskiy et al., 2021] and a powerful language model fine-tuned for vision-language reasoning tasks.
- InstructBlip-T5 [Dai et al., 2023], a model combining the T5 architecture [Raffel et al., 2020] for text processing with a highly efficient vision encoder, fine-tuned for instructional prompts and multimodal interactions.
- LLaVA-Next (Vicuna-7b) [Liu et al., 2024b], a VLM using Vicuna-7b-v1.5 [Zheng et al., 2024] and CLIP ViT-L/14 [Radford et al., 2021] as text and visual encoders, respectively. These are connected via simple projections.
- Qwen-VL-Chat [Bai et al., 2023], a 9B parameter model employing a cross-attention module to link an OpenClip ViT-bigG [Ilharco et al., 2021] vision encoder to a Qwen-7b [Bai et al., 2023] text backbone.
- Llava1.5-7b and 13-b [Liu et al., 2024a], a VLM employing a 7-billion parameter language model and advanced visual encoder, connected via efficient projections.
- InstructBlip-7b and 13b [Dai et al., 2023, 2024], a BLIP [Li et al., 2022] model fine-tuned using instruction tuning, using a 7-billion parameter language model and a high-resolution vision encoder for precise multimodal understanding.
- MoE-StableLM, MoE-Qwen, MoE-Phi2 [Lin et al., 2024], a mixture of experts (MoE) model combining StableLM architecture [Raffel et al., 2020] with multiple expert models for dynamic task specialization and improved performance.
- GPT-4o, a multimodal version of GPT-4 [OpenAI, 2023], incorporating optimized end-to-end multimodal encoding of images, text, and audio for improved multimodal task performance.
- Gemini-Flash [Gemini Team et al., 2023], a high-speed VLM combining the latest advancements in vision transformers [Dosovitskiy et al., 2021] and language models for rapid and accurate multimodal analysis.

Note that, for the last two models in this list, we are unable to provide any specific information regarding their respective architectures or training regimes, as this information has not been made publicly available.

## C.3 Prompts

We use the following general prompt template for our zero-shot experiments:

- T1 Prompt: This image contains a {shape} integrated into a background, where elements of the background contribute to forming the {shape}. Identify the {shape} that is represented in the image by choosing exclusively among the following options: {shape_options}, {background_classes}. Provide your response by stating only the single, most accurate class name that represents the {shape}. You have to respond with a single word.
- Texture Question Bias: This image contains a {shape} integrated into a background, where elements of the background contribute to forming the {shape}. Identify the background that is represented in the image by choosing exclusively among the following options: {shape_options}, {background_classes}. Provide your response by stating only the single, most accurate class name that represents the background. You have to respond with a single word.

where shape $\in$ {logo, shape, icon}for the dataset IllusionBench-LOGO, IllusionBench-IN and IllusionBench-CI respectively.

## C.4 Text Generation Hyperparameters

For all VLMs, we use full-precision weights (i.e., no quantization), generating responses using greedy decoding without sampling, and limit the maximum response length to 100 tokens.

## C.5   Zero-Shot Results By Class

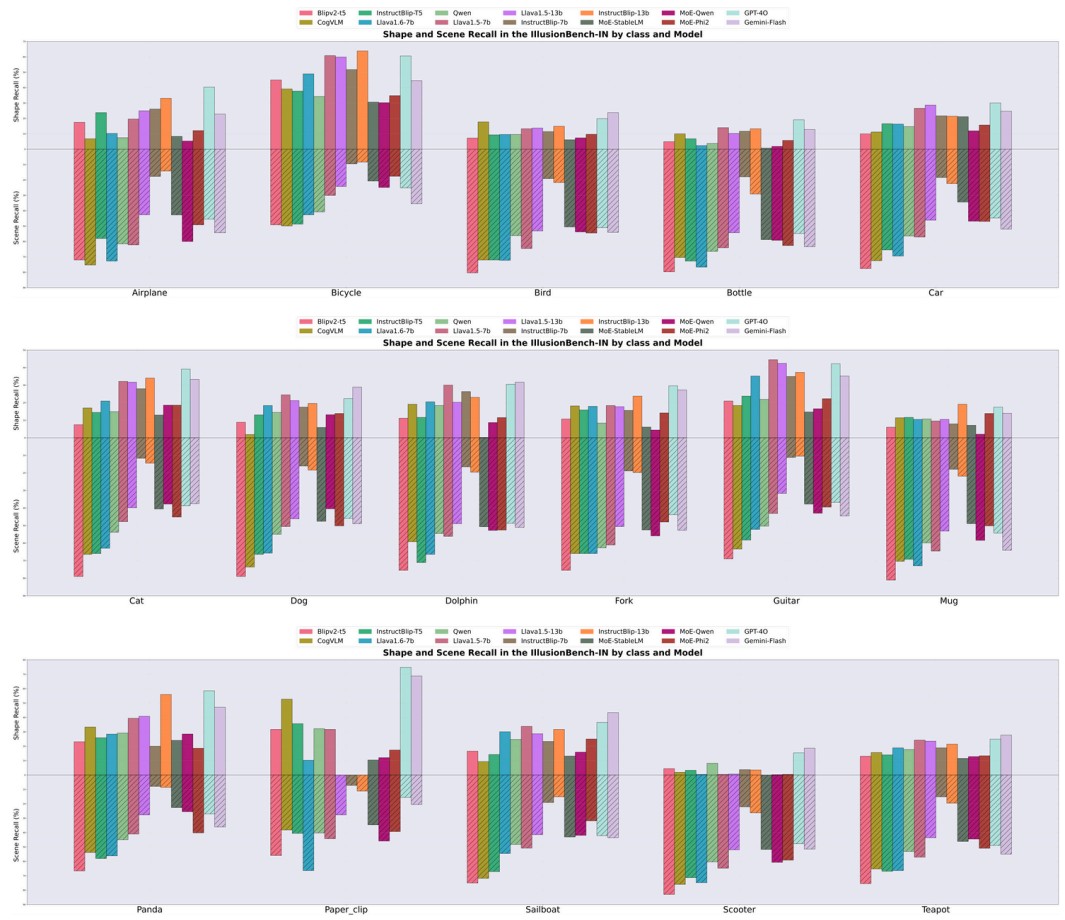

Figure 10: **Zero-shot results on** `IllusionBench-IN` **by class.** Zero-shot shape and scene recall of VLMs for each class in the `IllusionBench-IN` dataset.

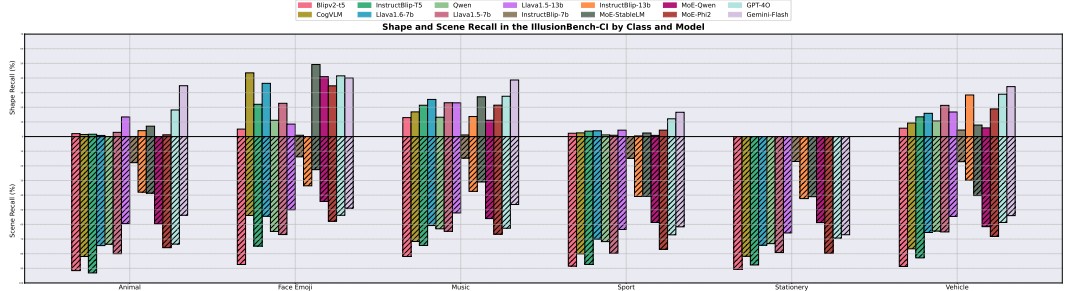

Figure 11: **Zero-shot results on** `IllusionBench-LOGO` **by class.** Zero-shot shape and scene recall of VLMs for each class in the `IllusionBench-LOGO` dataset.

Figure 12: **Zero-shot results on** `IllusionBench-ICON` **by class.** Zero-shot shape and scene recall of VLMs for each class in the `IllusionBench-ICON` dataset.

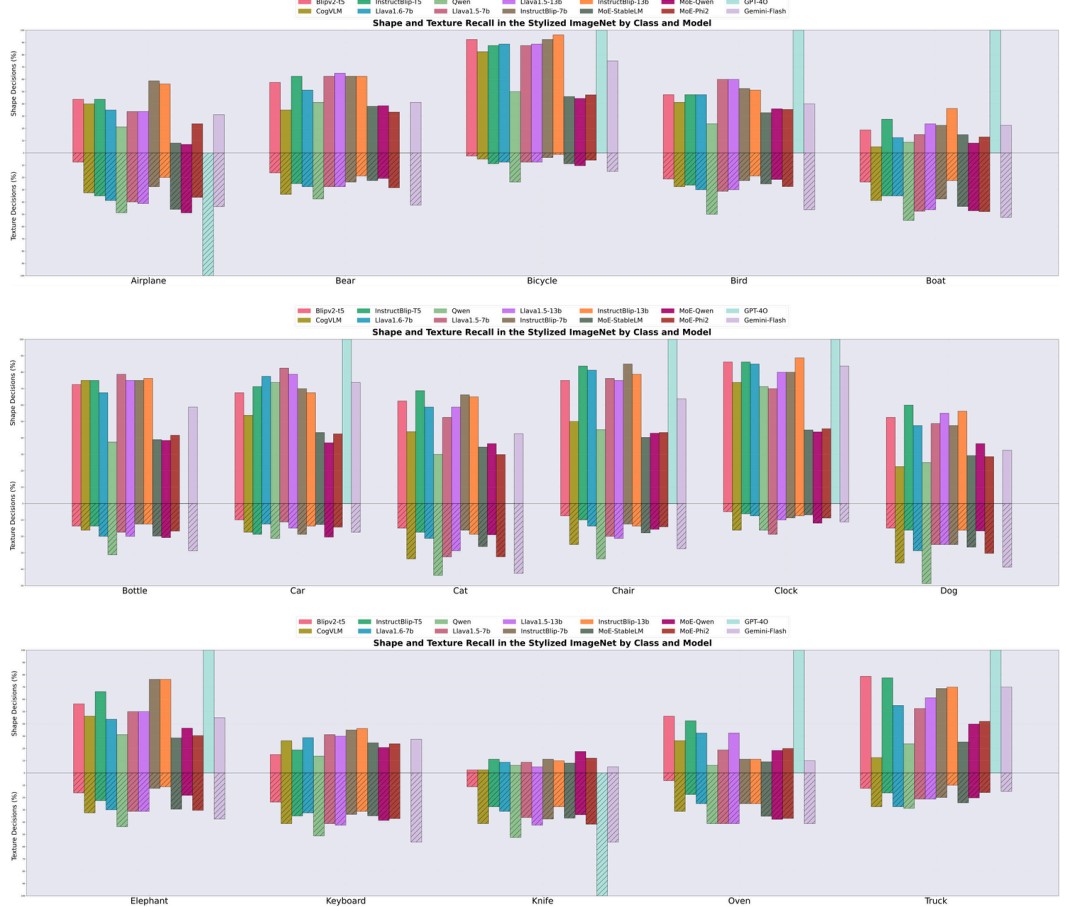

Figure 13: **Zero-shot results on Stylized ImageNet by class.** For comparison, we also report the zero-shot shape and texture bias of VLMs on the Stylized ImageNet dataset [Geirhos et al., 2019].

## D In-Context Learning Experiments Details

### D.1 In-Context Learning (ICL)

ICL is a method of adapting a model for an unseen task without any additional training or fine-tuning. Specifically, $n$-shot ICL consists of sequence of labelled demonstrations $\mathcal{C} = \{(x_{i_1}, y_{i_1}), \cdots, (x_{i_n}, y_{i_n})\}$. These are supplied to a model $p_{\boldsymbol{\theta}}(y|x)$ for an unseen task. The label corresponding to a test query $x_*$ is predicted through the predictive distribution of the model conditioned on the demonstration set $\mathcal{C}$ alongside an instruction $I$ for the new task:

$$p_{\boldsymbol{\theta}}(y|\mathcal{C}, I) = p_{\boldsymbol{\theta}}(y|x_{i_1}, y_{i_1}, \cdots x_{i_n}, y_{i_n}, I). \tag{1}$$

This learning method has proven to be an efficient and low-cost method for adapting LLMs to downstream tasks [Brown et al., 2020, Schick and Schütze, 2021, Winata et al., 2021, Liu et al., 2022]. The success of ICL for LLMs has led to recent research aiming to extend ICL to multi-modal models, where labeled demonstrations now contain interleaved image and text modalities [Alayrac et al., 2022, Bertini Baldassini et al., 2024, Zhao et al., 2023, Zong et al., 2024].

### D.2 ICL Further Experimental Details

Considering we restrict evaluations to classes recognised in a zero-shot manner, we use the following class counts: 10 for the IllusionBench-LOGO split, 14 for the IllusionBench-IN split, and 6 for the icons split, utilizing all 11 scenes of the dataset. To overcome ICL biases like majority voting and recency bias, each shape and scene class is represented at most once within the context, with no repetitions, and new demonstrations are randomly sampled for each test sample.

### D.3 Models Description

In our zero-shot experiments, we evaluate each of the following large vision language models (VLMs):

- LLaVA-Next (Vicuna-7b) [Liu et al., 2024b], a VLM operating at an input image resolution of $336^2$, using Vicuna-7b-v1.5 [Zheng et al., 2024] and CLIP ViT-L/14 [Radford et al., 2021] as text and visual encoders, respectively. These are connected via simple projections.
- Qwen-VL-Chat [Bai et al., 2023], a 9B parameter model with an input resolution of $448^2$, employing a cross-attention module to link an OpenClip ViT-bigG [Ilharco et al., 2021] vision encoder to a Qwen-7b [Bai et al., 2023] text backbone.
- Otter-MPT [Li et al., 2023a], a 9B parameter VLM based on the OpenFlamingo architecture [Awadalla et al., 2023], featuring an input image resolution of $224^2$ and utilizing LLaMA-7B [Touvron et al., 2023] and CLIP-ViT-L/14 as text and image backbones, respectively, connected through cross-attention.
- IDEFICS-9B-Instruct [Laurençon et al., 2024], an open-source reproduction of Flamingo [Alayrac et al., 2022], with an input image resolution of $224^2$, using cross-attention transformer blocks to connect LLaMA and OpenClip text and image backbones.
- MMICL-T5-XXL [Zhao et al., 2023], a 12B parameter model that employs a Q-former [Li et al., 2023b] to integrate language and image components within an InstructBlip-FLANT5-XXL [Dai et al., 2024] backbone. This model can handle complex prompts with interleaved text and images, allowing for text-image references through dummy demonstration tokens, and operates at an input image resolution of $224^2$.

### D.4 Prompts

We use the following general prompt template for our ICL experiments:

```
{TASK_INSTRUCTION}
{demonstration_image_1}
Answer: {demonstration_label_1}
{demonstration_image_2}
Answer: {demonstration_label_2}
.
.
.
{demonstration_image_n}
Answer: {demonstration_label_n}
{query_image}
Answer:
```

where `demonstration_image_i` and `demonstration_label_i` refer to the image and label for the $i$th demonstration used as the context for predicting the answer for the query image `query_image`. `TASK_INSTRUCTION` is the instruction used based on the prediction target and the dataset. We used the following `TASK_INSTRUCTION` prompts for predicting the shape, texture, and both the texture and shape simultaneously respectively:

```
# Predict shape
TASK_INSTRUCTION ='This image contains a {shape} integrated into a
background, where elements of the background contribute to forming
the image.
background options:  [{BG_OPTIONS}]
{shape} options:  [{SHAPE_OPTIONS}]
Identify the {shape} that is represented in the image by choosing
among the provided options.  Provide your response by stating only
the single, most accurate option that represents the {shape} in the
image.  You have to respond with a single word.'

# Predict texture
TASK_INSTRUCTION = 'This image contains a {shape} integrated into a
background, where elements of the background contribute to forming
the image.
background options:  [{BG_OPTIONS}]
{shape} options:  [{SHAPE_OPTIONS}]
Identify the background that is represented in the image by choosing
among the provided options.  Provide your response by stating only
the single, most accurate option that represents the background in
the image.  You have to respond with a single word.'

# Predict both texture and shape
TASK_INSTRUCTION = 'This image contains a {shape} integrated into a
background, where elements of the background contribute to forming
the image.
background options:  [{BG_OPTIONS}]
{shape} options:  [{SHAPE_OPTIONS}]
Identify BOTH the background AND the {shape} that are represented
in the image by choosing among the provided options.  Provide your
response by stating only the single, most accurate options that
represent the background and the {shape} in the image respectively.
You have to respond with two words, one predicting the background and
one predicting the {shape}'
```

where shape ∈ {logo, object, icon}for the dataset IllusionBench-LOGO, IllusionBench-IN and IllusionBench-CI respectively.

## D.5  Text Generation Hyperparameters

For all VLMs, we use full-precision weights (i.e., no quantization), generating responses using greedy decoding without sampling, and limit the maximum response length to 100 tokens.

## D.6  ICL Results: Exceptions

We list the exceptions to the general treneds reported in Section 5. We maintain the key takeaway headings and format in Section 5 and discuss key exceptions.

- *ICL does not mitigate tendency to predict scene over shape.* LLaVA on the task $\tau_C$ (along the first row) stands as an exception, where the model demonstrates low scene prediction accuracy and non-trivial performance shape accuracy on ICL2 and ICL4.
- *Context selection strategy effects prediction tasks differently.*
  - $\tau_C$ : For LLaVA, including the shape through ICL2 or ICL4 for 1 or 2 shots leads to a significant performance increase over all other models. This is especially evident for 1-shot, where we see high shape accuracy values of ICL2: 97.9% and ICL4: 99.9%. These high accuracy values indicate that the model exhibits a copying phenomenon [Bertini Baldassini et al., 2024], where for 1-shot, it simply copies the label from the ICL demonstration, which will have the same test label.

- $\tau_S$: QWEN shows an improvement in scene accuracy (for 4-shot, scene accuracies are ICL1: $51.4\%$ and ICL3: $88.1\%$) when the scene is included in the context. Additionally, LLaVA exhibits a similar copying phenomenon for scene prediction in ICL3 and ICL4 as discussed for $\tau_C$ but also shows some improvements over zero-shot for 2-shots.
- $\tau_{C,S}$: As an exception, OTTER and QWEN show a general increase in scene accuracy on $\tau_{C,S}$ compared to $\tau_S$, while their shape accuracy remains similar to $\tau_C$. This suggests that predicting both shape and scene and including demonstrations with such predictions can help these models better disentangle scene from shape. Again, we observe the copying mechanisms in LLaVA described for $\tau_C$ and $\tau_S$.

### D.7 Individual Dataset Splits ICL Results

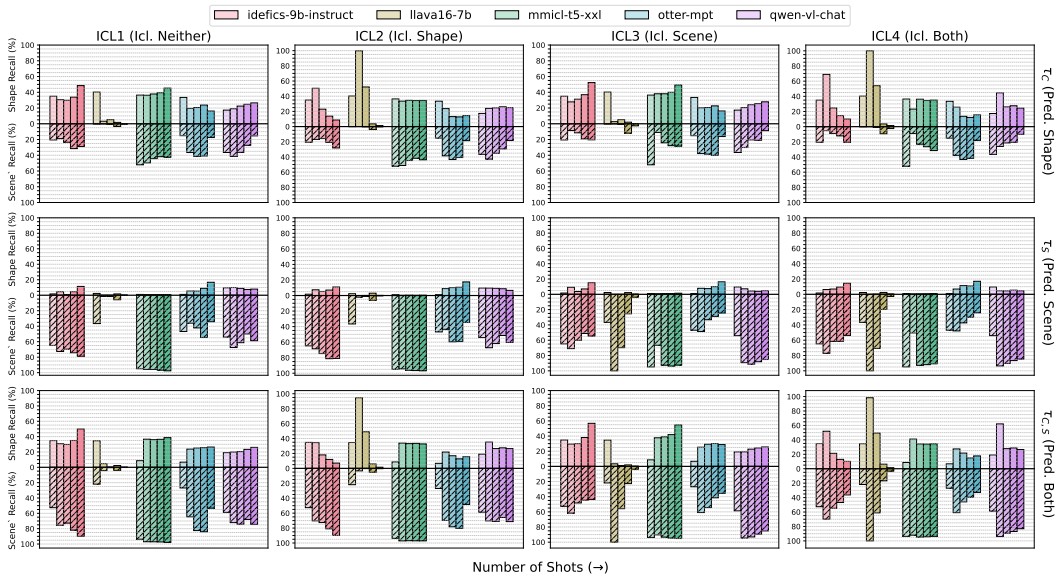

Figure 14: **ICL Results on `IllusionBench-LOGO`**. Few-shot shape and texture accuracy of VLMs on the `IllusionBench-LOGO` dataset across the different ICL learning tasks and the different prediction tasks.

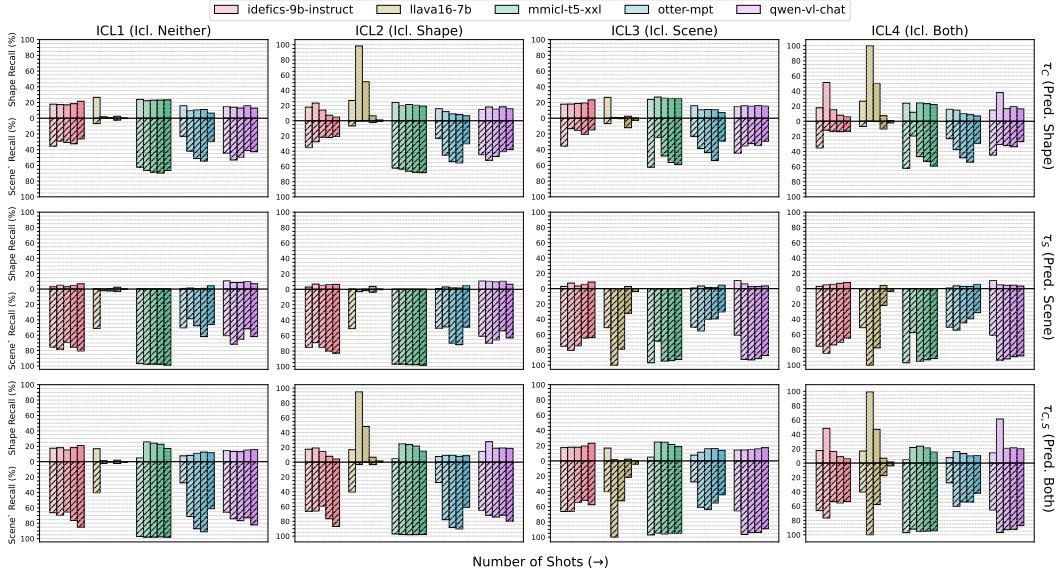

Figure 15: **ICL Results on `IllusionBench-IN`**. Few-shot shape and texture accuracy of VLMs on the `IllusionBench-IN` dataset across the different ICL learning tasks and the different prediction tasks.

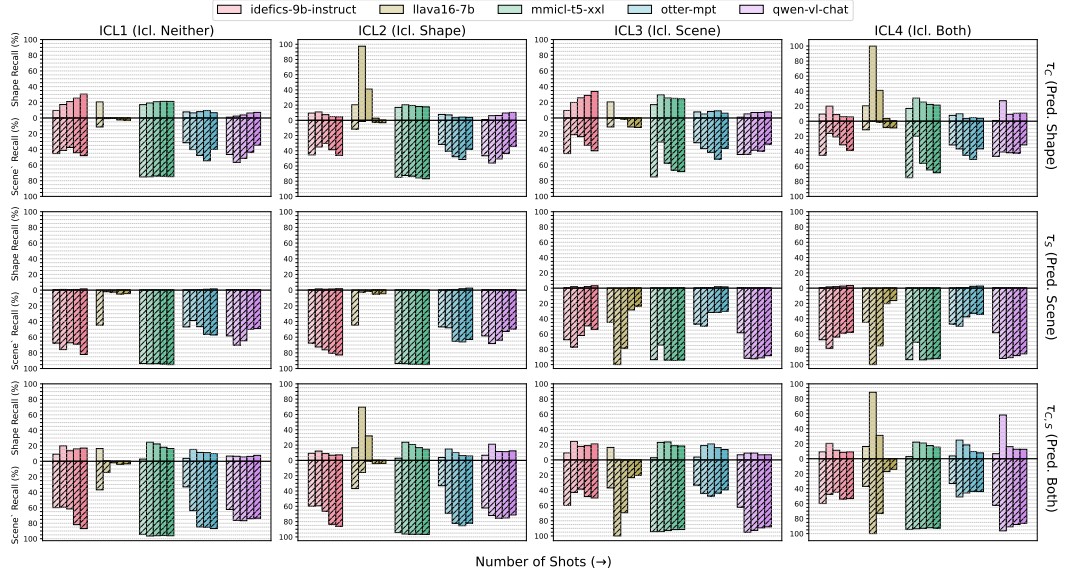

Figure 16: **ICL Results on `IllusionBench-ICON`.** Few-shot shape and texture accuracy of VLMs on the `IllusionBench-ICON` dataset across the different ICL learning tasks and the different prediction tasks.

## D.8 Responses From Low Performing Models

We often observe close to 0% shape accuracy of the `LLaVA` model on shape prediction tasks across all four ICL-constrained ICL prediction tasks when using a higher number of ICL demonstrations. Figure 17 illustrates three example responses from the `LLaVA` model using 4-shot ICL for ICL3, which includes the test query background in the ICL demonstrations. From the example model responses R1, R2, and R3, it is evident that the `LLaVA` model tends to produce descriptive and verbose responses. Specifically, it fails to be concise and accurate, unlike the other models we investigate that usually respond with a single class prediction even with more shots. This verbosity leads to poor accuracy as the model fails to adhere to the prompt instructions of predicting a single class, resulting in the test class rarely being included in the model's responses.

> - **R1**: The image shows a paper sculpture that resembles a stylized
> - **R2**:    The image shows a logo integrated into a background that features a mountainous landscape
> - **R3**: The image shows a beautiful natural scene with a large rock formation in the ocean

Figure 17: `LLaVA` **verbose responses.** Example responses from the `LLaVA` model for 4-shot shape prediction (T1) on the ICL3 learning task.

However, Figure 18 shows example responses from the `LLaVA` model on the same task and for the same test queries as in Figure 17 but using 2-shots. Observations from responses R1', R2', and R3' indicate that with fewer shots, the model is much more likely to produce single-class predictions or responses that are generally more concise and less descriptive. The differences observed with increasing numbers of shots suggest that `LLaVA`'s ability to correctly process and learn both the expected answer format and the task diminishes with a greater number of shots, highlighting its limitation as an in-context learner.

> - **R1'**: The logo in the image is Tesla.
> - **R2'**: The logo in the image is Starbucks.
> - **R3'**: Audi

Figure 18: `LLaVA` **concise responses.** Example responses from the `LLaVA` model for 2-shot shape prediction (T1) on the ICL3 learning task for the same test query as in Figure 17.

### D.9 ICL Prompt and Context Sensitivity

**Prompt Sensitivity.** We assess whether our ICL results are sensitive to the prompts used. We conduct ablations over four different prompt templates: the original template provided in Appendix D.4 and three additional variations. These variations include: (i) a simplified minimalistic prompt, (ii) the same simplified prompt but reversing the order in which the object and background options are presented, and (iii) a Llama-guard-style prompt [Inan et al., 2023] that explicitly indicates what the model should and should not focus on when making predictions. The specific prompt templates are as follows:

```
# Simplified prompt
TASK_INSTRUCTION = 'This image contains an object integrated into a
background, where elements of the background contribute to forming
the image.
background options:  [{BG_OPTIONS}]
{object} options:  [{OBJ_OPTIONS}]
Identify the object/background/object and background that are
represented in the image by choosing among the provided options.'

# Simplified prompt reverse
TASK_INSTRUCTION = 'This image contains a background with an
integrated {object}, where elements of the background contribute to
forming the image.
{object} options:  [{OBJ_OPTIONS}]
background options:  [{BG_OPTIONS}]
Identify the {object/background/object and background} that are
represented in the image by choosing among the provided options.'

# Llama-guard style
### Pay attention to:
 ONLY {the object/the background/BOTH the object and the background} that is represented
in the image by choosing among the provided icon options.TASK_INSTRUCTION =
'This image contains an {object} integrated into a background, where
elements of the background contribute to forming the image.
{object} options:  [{OBJ_OPTIONS}]
background options:  [{BG_OPTIONS}]
Identify the {object/background/object and background} that is
represented in the image by choosing among the provided options.
Provide your response by stating only the single, most accurate
option that represents the {object/background/object and background}
in the image.  You have to respond with a single word.

### Pay attention to:
   ONLY {the object/the background/BOTH the object and the background}
that is represented in the image by choosing among the provided icon
options.

### DO NOT:
 Focus on the {object/the background/IGNORE IN THIS CASE} of the
image.'
```

We report the mean shape and scene recall on the `IllusionBench-LOGO` dataset split, with error bars representing one standard error from the mean. The results are shown in Figure 19. Overall, we observe very little variation in shape and scene recall across models, tasks, and contexts. Significant variations, when present, occur only for LLava or Idefics models and are limited to cases with a small number of shots. These variations diminish as the number of in-context examples increases, suggesting that the results described in Section 5 are generally insensitive to the type of prompt used, particularly when a larger number of in-context examples are provided.

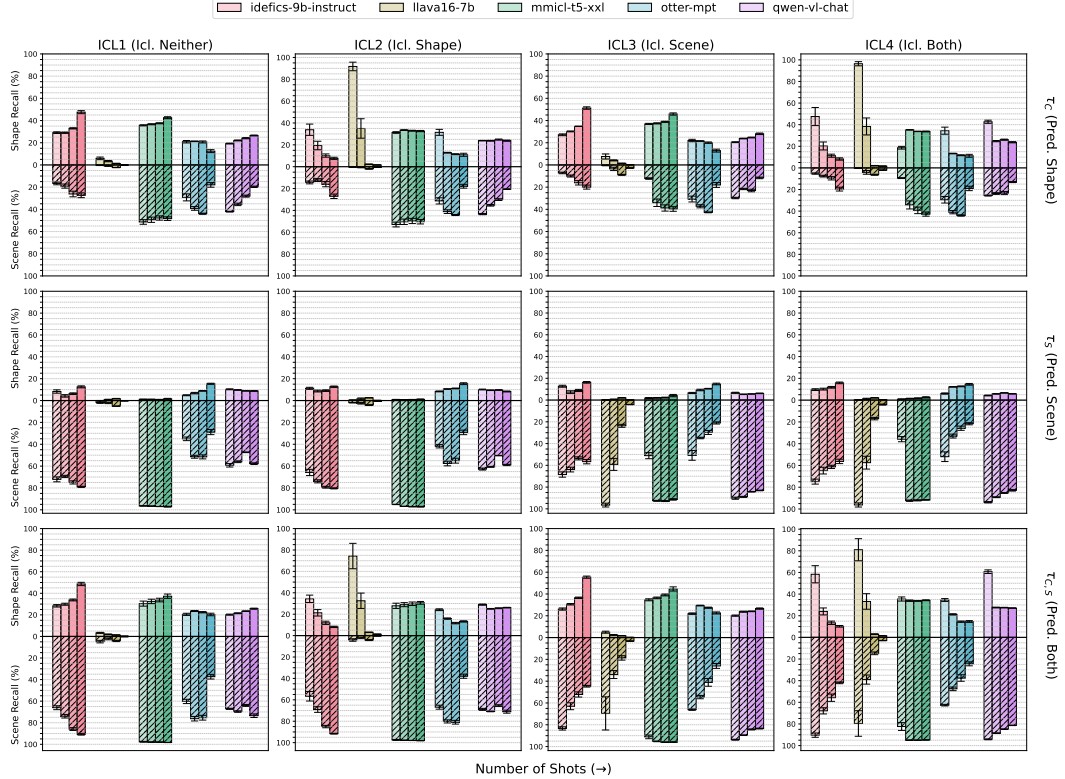

Figure 19: **Prompt sensitivity on** `IllusionBench-LOGO`. Mean shape and scene recall metrics with error bars representing one standard error from the mean across four different prompts used for ICL on the `IllusionBench-LOGO` dataset split.

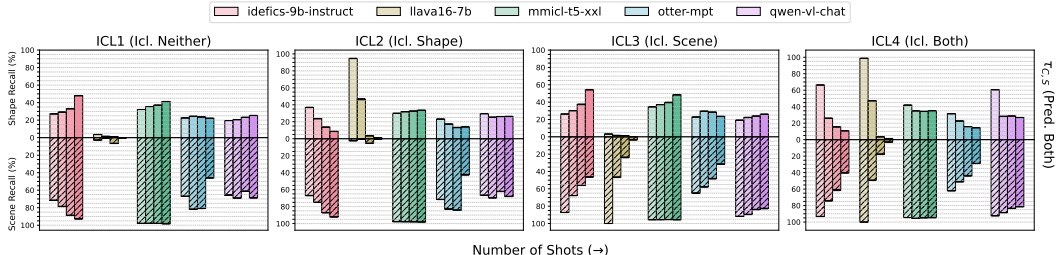

Figure 20: **Context ordering sensitivity on** `IllusionBench-LOGO`. Mean shape and scene recall metrics with error bars representing one standard error from the mean over three shuffled orders of the same context examples used for ICL on the `IllusionBench-LOGO` dataset split.

**Sensitivity to the order of in-context examples.** We also investigate the sensitivity of ICL results to the order of in-context examples. To assess this, we shuffle the context examples three times on the `IllusionBench-LOGO` when performing inference on task $\tau_{C,S}$. The results, shown in Figure 20, display very tight metrics with minimal variation in shape and scene recall, demonstrating that the results described in Section 5 are not sensitive to the ordering of the context examples.

# E   Domain Generalisation Experiments Details

## E.1   Background Details

Domain generalisation has been a challenging task for image recognition. Several methods have been developed to improve training strategies for better generalisability of early specialist visual models, which are also applicable to CLIP models. Data augmentation strategies such as MixUp [Yan et al., 2020] and RegMixUp [Pinto et al., 2022b] are known to improve generalisation capacity through

interpolation or extrapolation of data samples outside the training domain for diversity. GroupDRO [Sagawa et al., 2019] performs ERM with a re-weighting of classes with larger errors, making them more significant. VREx [Krueger et al., 2021] reduces differences in risk across training domains, which can decrease a model's sensitivity. Additionally, prompt learning, a promising approach for CLIP-style models, can also be leveraged for domain generalisation. Specifically, we adopt DPLCLIP [Zhang et al., 2023b], which trains a prompt generator during the training phase and infers unseen domains.

## E.2  Further Experiment Details

**CLIP Model**  For all experiments, the image encoder backbone of CLIP model is a ResNet50 [He et al., 2016]. For full-parameter fine-tuning, we train the whole image encoder, whereas for linear probing we only train the projection layer. The inferent prompt template for all methods is ``A photo of [Class name]''.

**Training Hyperparameters**  For all experiments, we use a batch size of 32 and the Adam optimiser [Kingma and Ba, 2014] with a learning rate of 5e-5. For full parameter fine-tuning, we train the model for 1000 steps, and for linear probing, we train the model for 800 steps. For MixUp [Yan et al., 2020] and RegMixUp [Pinto et al., 2022b], the alpha and beta are both set to 0.2. For GroupDRO [Sagawa et al., 2019], the eta is set to 1e-2. For VREx [Krueger et al., 2021], the penalty weight is set to 1.0. For DPLCLIP [Zhang et al., 2023b], the number of domain tokens is 16.

## F  Compute Resources

All experiments are performed on our internal cluster.

**Resources for image generation**  For the Image generation, we used three A40 GPUs with 45 GB RAM with around 65h to generate all of the images in the dataset.

**Resources for zero-shot experiments**  For the zero-shot experiments, we used eight A40 GPUs with 45 GB RAM for around 250h total to cover all Zero-shot experiments experiments.

**Resources for in-context learning experiments**  We perform ICL inference using 8 A40 GPUs with 45GB RAM for around 168h total to cover all ICL experimental settings.

**Resources for domain generalisation experiments**  For each fine-tuning CLIP we use a single A40 GPUs with 45GB RAM for an hour on average for full parameter fine-tuning and half an hour for linear probing.

