# OpenReview forum: "Hidden in Plain Sight: Evaluating Abstract Shape Recognition in Vision-Language Models"
_NeurIPS.cc/2024/Datasets_and_Benchmarks_Track — NeurIPS 2024 Track Datasets and Benchmarks Poster_

### Official Review · Reviewer_F7Nb · 2024-07-16
**Benchmark for evaluating VLMs on Abstract Shape Recognition**

**Rating:** 7
**Confidence:** 1
**Correctness:** The claims made sound correct.
**Clarity:** The paper is overall well written.

**Review:**

1. The paper is overall well written, claims made are coherent, details are clear.
2. The paper performed thorough evaluation and analysis. The findings are grounded with experiment results.
3. The paper lacks a good discussion of prior works, e.g., HallusionBench. Therefore, the originality of the work couldn't be assessed correctly. Nonetheless, the contribution of the work still seem significant. The findings of the papers would foster future research.

**Strengths:**

1. The paper presents a dataset to evaluate VLMs' capabilities of recognizing abstract shapes.
2. The paper thoroughly studies zero-shot recognition, few-shot learning of VLMs, and domain generalization capabilities.

**Additional Feedback:**

None

**Documentation:**

The paper presents sufficient detail on data collection and organization.

**Ethics:**

There is no ethical concern.

**Limitations:**

The paper does not discuss limitations, however, it discusses the societal impact.

**Opportunities For Improvement:**

The proposed dataset name is IllusionBench which is pretty closer to HallusionBench. Is there a better way to name the dataset?

**Relation To Prior Work:**

I am not sure but it seems like the paper didn't discuss all prior works well. For example, the popular HallusionBench is overlooked and it seems like that benchmark is pretty close to this work.

**Summary And Contributions:**

The paper presents IllusionBench, a collection of 3 datasets to evaluate shape recognition capabilities of vision language models. The work shows that while human identify these shapes with high accuracy, the state-of-the-art vision-language models fail to identify the shapes in these scenes zero-shot and tend to focus on scene elements instead. The authors further observed that in-context learning didn't help significantly, however, the CLIP models can be fine-tuned to recognize the shapes and generalize to new scene domains.

---

> ### Author Rebuttal · Authors · 2024-08-17
>
> We thank the reviewer for their positive assessment of our work, particularly in recognizing its **clearness and coherence, thorough analysis, claims/findings supported by experimental results**, and that the **contribution** of our work seems **significant**.
>
> **1. HallusionBench**. We thank the reviewer for drawing our attention to the HallusionBench benchmark, and agree that this name is similar enough to “IllusionBench” that it could possibly lead to confusion between our two benchmarks. We will discuss HallusionBench in our updated draft to better distinguish it from our benchmark and more clearly scope the novelty of our work. To this end, we summarize the relationship between each work below:
>
> **HallusionBench**:
> - The goal of HallusionBench is to provide a **diagnostic tool** to distinguish whether a VLM **failure is caused by the language or visual understanding part** of a multi-modal model. The authors refer to failures caused by an erroneous visual processing of the input as visual illusions.
> - Additionally, the authors include a split of their dataset by collecting known illusory patterns that typically **confuse humans** (e.g., determining the relative size of two objects given they are placed in different locations over a grid that is perspectically distorted, or surrounded by objects of different sizes) and **checking if they confuse VLMs too**.
>
> **IllusionBench**:
> - Our goal is to create a benchmark that studies abstract shape recognition using “illusion”-like imagery (similar to pareidolia), where **samples are clearly human-perceptible**. We demonstrate that humans can easily perceive the shapes contained in the images (as observed in their high shape recognition accuracy).
> - We study the specific failure case in which **VLMs fail** to recognize the shape of objects when they are represented by the composition of other scene elements.
> Thus, **as acknowledged by the reviewer, our contribution is quite distinct** from that of HallusionBench. However, we agree that it would be helpful to reference HallusionBench in our updated draft to more clearly distinguish it and resolve any potential confusion due to their similar names.
>
> **2. Name Change Suggestion.** As we have already made the dataset available in HuggingFace Datasets (as recommended in the checklist and call for papers) using this name, this may be a bit tricky. However, we welcome any suggestions from the reviewer regarding any additional colloquial terms we could use to refer to our dataset to make ours more easily distinguished from HallusionBench. For instance, we would be open to renaming the splits from “IllusionBench-{IN,LOGO,ICON}” to “IB-{IN,LOGO,ICON}”, so that they can be more easily distinguished from HallusionBench. We are open to any other suggestion.
>
> In sum, we appreciate the reviewer’s positive feedback, and we hope that our responses will be satisfactory. We are open to any further suggestions (e.g., regarding colloquial renaming of our benchmark or constituent splits internally throughout the paper), and will remain available to respond to any further questions, comments, or suggestions.

---

### Official Review · Reviewer_cHDM · 2024-07-17
**A shape recognition benchmark for VLMs**

**Rating:** 5
**Confidence:** 5
**Correctness:** Some of the analysis may be questiona…

**Review:**

The proposed benchmark is a very interesting submission that overhauls the dated cue-conflict/Stylize-ImageNet datasets used in most publications to study shape recognition. I particularly like that this benchmark does not introduce a trade-off (e.g., shape vs. texture as in previous works) but allows the classification of shapes or scenes independently. As such this dataset (and novel benchmarking method) is better suited for modern research on shape recognition. Another important strength of this submission is the analysis. It dives deep into interesting aspects of VLMs (zero-shot, in-context learning, and generalization). While I think that the dataset is very good I do have some issues with the analysis and drawn conclusions from it (see below) that currently prevent me from arguing for acceptance. I hope that these can be resolved during the rebuttal, so I can increase my score accordingly for this otherwise strong submission.

**Strengths:**

- Well written and mostly easily understandable
- Overhauls existing shape benchmarks in a novel and more sophisticated manner paving the way for better insights
- A thorough benchmark on many recent VLMs is provided in multiple scenarios
- This is the only paper in my batch that delivered all the required metadata for the D&B track.

**Additional Feedback:**

- You already have many models in your analysis so I do not require this but given that InternVL seems to have stood out in the experiments of (Gavrikov et al., 2024) it might be an interesting data point.

**Clarity:**

The paper is well-written however I'd recommend polishing it a bit more to fix a few issues:
- Reference in L124 missing
- Appendix L592: "HamAn" annotators
- Legend in Fig. 5
- Fig. 8 in the appendix is not readable

**Documentation:**

The dataset is overall well documented and fulfills all requirements.

**Ethics:**

No concerns.

**Limitations:**

Limitations were adequately addressed, but I suggest discussing the human evaluation (see above).

**Opportunities For Improvement:**

- **Human study/framing of the problem**:  A lot of these illusions do not immediately stand out. In some sense, you can often only find them if you know that you are looking for one. Looking at multiple instances certainly has "primed" some human reviewers and the resulting accuracy might be overreporting the baseline. I don't think that this needs to be fixed, but it would be good to point this out in the limitations.
- **VLM Prompting**: Our lab also experimented with ControlNet illusions and VLMs and we found that the prompt heavily influences the output. Even semantically identical prompts (to humans) sometimes significantly differed in their outputs (i.e., models suddenly no longer saw illusions). It is currently not clear if the poor performance reported in this paper is indeed due to poor representations or simply ineffective prompts. It would be helpful to 1) ablate some prompts here and 2) shuffle prompt options. I am particularly worried about this, as some of the results on ICON are below random guesses which may be a result of "bad" prompting.
- **VLM Baseline:** Some of the illusions require fine "world knowledge" - i.e., I expect all models to detect shapes of cats but not necessarily be able to detect a Nike swoosh. I am missing a baseline/sanity-check that shows that VLMs can classify the raw shape images or scenarios (before turning them into illusions).
- **Response processing**: Currently it is not well documented how the accuracy is computed. The authors instruct models to respond only with the correct option but the model can ignore this. What happens then? How is the response "normalized" for computation of the accuracy?
- **VLM Parameters are not documented**: I am missing statements of how VLMs were loaded (precision, quantization) and stochastic token sampling (temperature, top_k etc.) was utilized (which is often the default option). This could heavily affect results.
- **Codebase**: Given that this seems to be primarily framed as a benchmark I also checked the code: currently it is poorly documented, very messy, and contains multiple hard-coded paths. It needs to be heavily polished to be useful.


Nitpicks:
- It was shown a few times now that shape bias (as measured on cue-conflict) is not a good indicator of generalization (L31) [1, 2].
- For CLIP the prompt shown in the paper does not seem to end with a ".". This can decrease accuracy (at least on OpenAI models)

[1] Mummadi et al., "Does enhanced shape bias improve neural network robustness to common corruptions?", ICRL 2021.

[2] Gavrikov et al., "Can Biases in ImageNet Models Explain Generalization?", CVPR 2024.

**Relation To Prior Work:**

Clearly discussed and sufficient.

**Summary And Contributions:**

The authors propose IlusionBench (in 3 variants) to test the shape recognition capabilities of vision language models. The dataset consists of "shapes" in the form of black images "hidden" in images of scenes via ControlNet. The authors pose the detection as a classification problem (of scenes or shapes). Additionally to the dataset, they provide an analysis of common VLMs in a zero-shot fashion, showing that many VLMs struggle to detect the hidden shapes that contrast previous findings. Additionally, the authors evaluate if in-context learning is helpful but do not find evidence of that. Lastly, they benchmark if invariant representations across domains be learned.

---

> ### Author Rebuttal · Authors · 2024-08-17
>
> We thank the reviewer for indicating that our work produces an **interesting dataset that is better suited to the modern capabilities of LLMs**, avoiding the previous trade-off between shape and texture understanding. We are happy to hear that they find our work **well written, understandable and providing a deep and comprehensive analysis of zero-shot prediction, in-context learning and domain generalization**. We hope our response will thoroughly address the reviewer’s concerns:
>
> 1. **Human evaluations formulation.**  The primary objective of the annotation was merely to confirm that the shapes are present in the generated images. The objective was *validating the quality of synthetic data and confirming that abstract shapes are human-perceptible,* not to provide a direct apples-to-apples comparison between human annotators and VLMs, as it is true that annotators are able to see several labeled images before providing their own annotations, whereas VLMs are not (at least in zero-shot evaluation). We have tried to emphasize this by naming the relevant paragraph **“Validating Dataset Quality” in section 3.2**.
> However, we appreciate the reviewers' insight regarding the limitations **if the goal is to evaluate human performance** on this task. Indeed, this priming phenomenon likely affects human performance in detecting shapes, and we will discuss it thoroughly in the revised manuscript (in section 3.2). Specifically, we will elaborate on how priming might simplify the detection of targeted shapes in more challenging samples.
>
> **2. Prompt Sensitivity:** We agree that the prompt could affect the performance. To more extensively support the claims made in our work, we re-run the experiments again using different prompts. Due to the computational cost of running these models and the limited time of the rebuttal, we perform this experiment only on a subset of models for all experimental settings. We will continue running the rest of the experiments to comprehensively update the final draft, and post them in a follow-up comment. **The currently available results do not change our findings.**
>
> **2.1. ICL Prompting**: For ICL, we use 3 additional prompts (see pdf attached) that significantly differ from the original one (see Fig 3).
> Across 4 different prompts, the variance is tight (error bars), except for the Llava models for a small amount of shots. However, when the number of shots is as low as 1 or 2, the model is not actually learning to perform the task but mostly **copying the labels seen in the context** to the output. As the amount of ICL samples increases (i.e., as other labels are included), the performance and the error bars reduce, indicating a **very limited ability to understand both shape and scenes**.  For all experiments we will include error bars. To also address the reviewer's concern about the order of the samples, we shuffle the context several times at random and report the average results over 3 shuffles, yielding to extremely tight variance (see Fig 2). Upon inspection, sub-random results seem to be due to models hallucinating. For example, Llava babbles a lot more wth increasing context and breaks with 8-shots, indicating long context is still hard to handle for VLMs.
>
> **2.2 CLIP Prompting for Zero-Shot:** We ablate several prompts present in the list of prompts from [1], including a version of the prompts including ‘.’. The presence of ‘.’ in the prompt actually degrades the accuracy of the version of CLIP we use, which differs from the original CLIP release for which this phenomenon was observed. Although a few prompting strategies are more effective than the one reported in the main paper, we observe the **overall ability to recognise the shape remains extremely limited.**
> [1] https://arxiv.org/abs/2103.00020
>
> **3. VLMs require world knowledge.** Thank you for bringing up this important point. We did take this into consideration. We have conducted evaluations on samples that the VLM can recognize their corresponding conditioning shapes (the binary shapes). We will emphasize this point in a few locations in the paper as well to reduce confusion and make it clear to all readers that **models are only evaluated in contexts where they have demonstrated the necessary world knowledge of the shapes in our dataset**.
>
> **4. How is the answer processed to extract the evaluation merics?** We summarize this procedure in lines 131-137 – specifically, we check whether the labels appear in the generated text. To clarify, we simply check if a normalized (lowercase, without punctuation) variant of the class to be considered  (shape name, scene type or both depending on the task) is present in the answer. We will add this additional clarification to the draft as well.
>
> **5. VLMs Params not Documented.** Thank you for drawing attention to this point, which we agree could indeed have an important impact on results. We do **not quantize models, and use floats at the maximum precision available** at the respective open-weights repositories. For each vision-language model, we use the model’s **default** HuggingFace Transformers text generation hyperparameters for all experiments (generally, greedy decoding without sampling), only changing the maximum response length to 100 tokens for each model. We will clarify both points in the final draft.
>
> **6. Refactor the code:** We agree the codebase can be improved in terms of documentation, clarity, modularity, and generality (e.g., no hard-coded paths). We will refactor it to correct these issues, so that users can easily adapt our code for their evaluations.
>
> We appreciate the reviewer’s stated openness to potentially increasing their score and recommending acceptance based on our rebuttal. We hope this rebuttal addresses the reviewer’s concerns, and that the score could be correspondingly updated. Otherwise, we remain available to respond to any further questions and provide further clarification as requested.

---

> > ### Comment · Reviewer_cHDM · 2024-08-23
> >
> > I thank the authors for the rebuttal and have the following follow-up remarks:
> >
> > > 1. [...]  we will discuss it thoroughly in the revised manuscript (in section 3.2). Specifically, we will elaborate on how priming might simplify the detection of targeted shapes in more challenging samples.
> >
> > It would be great to show the actual changes that the authors intend to make.
> >
> > > 2. Prompt Sensitivity: We agree that the prompt could affect the performance. [...] Due to the computational cost of running these models and the limited time of the rebuttal, we perform this experiment only on a subset of models for all experimental settings. We will continue running the rest of the experiments to comprehensively update the final draft, and post them in a follow-up comment. The currently available results do not change our findings.
> >
> > It is not necessary nor sane to repeat all experiments on all models. Some kind of baseline experiment on a few models would suffice. However, please do provide actual numbers.
> >
> > > 3. VLMs require world knowledge. [..]
> >
> > This is still not clear to me. Does this mean that all VLMs achieve 100% accuracy on the raw shapes?
> >
> > > 5. VLMs Params not Documented. [...] (generally, greedy decoding without sampling)
> >
> > "Generally" sounds as if stochastic sampling was used in some cases. Which models were those, and how did this impact their performance?

---

> > > ### Author Response · Authors · 2024-08-28
> > >
> > > # Post 1
> > >
> > > **Overview of Response:** We thank the reviewer for their feedback to our rebuttal. We will spread our response across 6 posts (due to character limits): the first two posts (including this one) will respond to the high-level concerns raised by the reviewer, and the remaining 4 posts will contain tables reporting our results on prompt sensitivity experiments.
> > >
> > > **Human Evaluation and Priming:** Naturally, we would be happy to share our intended changes. While we are limited to uploading only a single-page PDF and thus cannot attach a full revised manuscript, _we plan to add the following text to the end of Section 3.2 (line 129):_
> > >
> > > > Note that human annotator accuracies are only intended to validate the quality of the generated dataset and confirm that the resulting abstract shapes are indeed human-perceptible, and are _not_ intended for direct comparison with VLM performance, as there are a few fundamental differences in how annotators and VLMs are tested. For instance, where VLMs do not know the purpose or structure of the task beyond what is included in the prompt, annotators are shown onboarding materials describing the task including 5 pre-annotated examples to teach them the task, and further receive feedback on their performance for 10 additional test images (in order to ensure that they understand the task). As such, annotator performance is more similar to few-shot evaluation rather than zero-shot, with the additional caveat that annotators are provided with much more information and context about the task they are performing than could be included in prompts (e.g., how the dataset is constructed and why, individualized feedback on the initial 10 test images, etc.), effectively “priming” them to recognize shapes where they might otherwise have perceived only the background scene.
> > >
> > > **Prompt Sensitivity:** Before providing further experimental evidence, we would like to bring the attention of the reviewer to the [PDF attached in our initial rebuttal](https://openreview.net/attachment?id=kjS6oowJ6N&name=pdf) that already provides baseline experiments on a few models. Based on that PDF, one can already conclude the following:
> > > - CLIP zero-shot experiments (Table 1 of the PDF): although prompting affects the performance, the overall shape detection ability remains poor.
> > > - ICL (Figures 2 and 3 of the PDF): although prompting can change the performance the shape detection ability remains poor and the standard deviation (shown as error bars) is tight.
> > >
> > > (Please refer to our earlier post for a more detailed discussion of these results.)
> > >
> > > In order to provide additional evidence that prompt sensitivity is not the primary cause of models’ poor performance at shape recognition, we perform additional experiments using different prompts for (1) all VLMs on the LOGO split in the ICL setting, across all 4 ICL conditions (where the context contains either the target shape, target background, both, or neither) and all 3 prompting tasks (for shape, background, or both); and (2) a subset of the VLMs on the LOGO split in the zero-shot setting (as the remaining splits and models are still being evaluated, and all results will be included in the final draft). We report the mean and standard deviation across the 4 prompts in the following 4 posts (due to character limits).
> > >
> > > In these experiments, we find that the performance using these different prompts are nearly identical to those reported in the main paper, demonstrating that these changes to the prompts do not yield satisfactory shape detection performance.
> > >
> > > **VLMs require world knowledge: Does this mean that all VLMs achieve 100% accuracy on the raw shapes?**
> > > _The normalized accuracy reported in the main paper (Figure 3) is obtained by averaging results for each VLM exclusively on samples obtained from raw shapes that the VLM can recognise in a zero-shot setting._ (For clarity, we will add the preceding italicized sentence to the main paper.) For example, from the 21 total classes and 39 respective conditioning images in the full LOGO dataset, consider a random subset of 5 LOGO classes and 2 conditioning images for each of these classes, and suppose a given VLM successfully classifies only 7/10 of these conditioning images. In this case, the VLM would be evaluated only on samples generated using those 7 conditioning images, and not the 3 others.
> > >
> > > Note that, in addition to this normalized accuracy reported in Figure 3, we also report per-class accuracies averaged across images generated using _all_ conditioning images for each class (not only those that are recognized by any given VLM) in Appendix B.4 (Figure 11).

---

> > > ### Author Response · Authors · 2024-08-28
> > >
> > > # Post 2
> > >
> > > **Impact of Text-Generation Hyperparameters:** We thank the reviewer for clarifying this important point. The reason we said “generally” is because CLIP (as used in our domain generalization experiments) is a contrastive vision-language encoder, meaning that it cannot, by default, be used to generate text in the same way as the generative vision-language models (VLMs). We confirm that, for all generative VLMs on all zero-shot and ICL experiments, each one uses greedy decoding without sampling.
> > >
> > > However, to further examine this important question and make our evaluation more exhaustive, we have performed again zero-shot experiments on LLaVa-1.5-7B, InstructBLIP-7B on the LOGOs split, using beam search (num_beams=5 and do_sample=False) and multinomial sampling (num_beams=1 and do_sample=True). As these results indicate, the decoding strategy has only a very small effect on the performance.
> > >
> > > | Models/Decoding | Greedy | Beam | Multinomial (mean ± std) |
> > > |---|---|---|---|
> > > | LLaVa-1.5-7B| 24.93 |24.98 |24.72 ± 0.23  |
> > > | InstructBLIP-7B| 25.42 |25.29 |25.31 ± 0.45 |
> > >
> > > We will include these experiments and additional ablations on the decoding hyperparameters for zero-shot and ICL models in the final draft.
> > >
> > > **Conclusion:** We hope we have adequately resolved each of the reviewer’s concerns. Otherwise, we remain available to provide additional clarifications and respond to any further questions.

---

> > > ### Author Response · Authors · 2024-08-28
> > >
> > > # Post 3
> > >
> > > ICL, LOGO, Task: Predict Shape
> > > | Context Contains | Model | Shot | Shape Accuracy (Mean ± SE) | Background Accuracy (Mean ± SE) |
> > > |:----------------|:------|:-----|:-------------------------:|:--------------------------:|
> > > | neither | otter-mpt | 1 | 20.93 ± 1.10 | 29.23 ± 3.15 |
> > > | neither | otter-mpt | 2 | 21.28 ± 0.25 | 39.19 ± 1.66 |
> > > | neither | otter-mpt | 4 | 20.71 ± 0.90 | 43.96 ± 0.50 |
> > > | neither | otter-mpt | 8 | 12.44 ± 1.33 | 17.96 ± 2.09 |
> > > | neither | llava16-7b | 1 | 5.94 ± 1.17 | 0.38 ± 0.19 |
> > > | neither | llava16-7b | 2 | 3.44 ± 0.58 | 0.94 ± 0.33 |
> > > | neither | llava16-7b | 4 | 1.27 ± 0.26 | 2.37 ± 0.03 |
> > > | neither | llava16-7b | 8 | 0.20 ± 0.06 | 0.36 ± 0.07 |
> > > | neither | mmicl-t5-xxl | 1 | 35.68 ± 0.50 | 51.37 ± 2.08 |
> > > | neither | mmicl-t5-xxl | 2 | 36.64 ± 0.46 | 49.37 ± 2.38 |
> > > | neither | mmicl-t5-xxl | 4 | 37.53 ± 0.45 | 47.80 ± 1.91 |
> > > | neither | mmicl-t5-xxl | 8 | 42.55 ± 0.88 | 48.38 ± 1.62 |
> > > | neither | qwen-vl-chat | 1 | 19.27 ± 0.37 | 42.06 ± 0.35 |
> > > | neither | qwen-vl-chat | 2 | 22.07 ± 0.41 | 35.53 ± 1.11 |
> > > | neither | qwen-vl-chat | 4 | 24.09 ± 0.53 | 28.08 ± 0.91 |
> > > | neither | qwen-vl-chat | 8 | 26.54 ± 0.39 | 19.71 ± 0.61 |
> > > | neither | idefics-9b-instruct | 1 | 29.09 ± 0.74 | 16.68 ± 1.21 |
> > > | neither | idefics-9b-instruct | 2 | 29.01 ± 0.57 | 19.06 ± 1.86 |
> > > | neither | idefics-9b-instruct | 4 | 32.97 ± 0.53 | 26.16 ± 2.55 |
> > > | neither | idefics-9b-instruct | 8 | 47.56 ± 1.36 | 27.31 ± 2.16 |
> > > | shape | otter-mpt | 1 | 31.44 ± 2.73 | 31.29 ± 2.83 |
> > > | shape | otter-mpt | 2 | 12.83 ± 0.35 | 41.13 ± 1.67 |
> > > | shape | otter-mpt | 4 | 11.44 ± 0.39 | 44.26 ± 0.46 |
> > > | shape | otter-mpt | 8 | 10.91 ± 1.40 | 17.86 ± 1.75 |
> > > | shape | llava16-7b | 1 | 91.76 ± 3.92 | 0.30 ± 0.16 |
> > > | shape | llava16-7b | 2 | 35.00 ± 8.90 | 0.76 ± 0.23 |
> > > | shape | llava16-7b | 4 | 2.24 ± 0.19 | 2.03 ± 0.24 |
> > > | shape | llava16-7b | 8 | 1.61 ± 0.17 | 0.56 ± 0.05 |
> > > | shape | mmicl-t5-xxl | 1 | 31.30 ± 0.73 | 52.94 ± 2.19 |
> > > | shape | mmicl-t5-xxl | 2 | 33.49 ± 0.34 | 50.43 ± 2.48 |
> > > | shape | mmicl-t5-xxl | 4 | 32.95 ± 0.41 | 49.68 ± 2.39 |
> > > | shape | mmicl-t5-xxl | 8 | 32.64 ± 0.54 | 50.37 ± 2.04 |
> > > | shape | qwen-vl-chat | 1 | 23.85 ± 0.21 | 43.35 ± 0.56 |
> > > | shape | qwen-vl-chat | 2 | 23.76 ± 0.36 | 35.32 ± 0.73 |
> > > | shape | qwen-vl-chat | 4 | 24.82 ± 0.49 | 30.11 ± 0.79 |
> > > | shape | qwen-vl-chat | 8 | 23.82 ± 0.53 | 20.56 ± 0.31 |
> > > | shape | idefics-9b-instruct | 1 | 33.84 ± 5.16 | 14.41 ± 1.28 |
> > > | shape | idefics-9b-instruct | 2 | 19.41 ± 3.90 | 12.25 ± 1.33 |
> > > | shape | idefics-9b-instruct | 4 | 10.19 ± 1.49 | 16.00 ± 2.38 |
> > > | shape | idefics-9b-instruct | 8 | 7.67 ± 0.98 | 27.07 ± 2.04 |
> > > | bkg | otter-mpt | 1 | 22.03 ± 1.06 | 30.73 ± 2.63 |
> > > | bkg | otter-mpt | 2 | 21.54 ± 0.25 | 37.25 ± 1.39 |
> > > | bkg | otter-mpt | 4 | 20.07 ± 0.62 | 42.73 ± 0.36 |
> > > | bkg | otter-mpt | 8 | 12.89 ± 1.29 | 18.05 ± 2.19 |
> > > | bkg | llava16-7b | 1 | 7.61 ± 2.27 | 0.48 ± 0.27 |
> > > | bkg | llava16-7b | 2 | 4.11 ± 0.47 | 3.58 ± 1.49 |
> > > | bkg | llava16-7b | 4 | 1.45 ± 0.17 | 8.91 ± 0.46 |
> > > | bkg | llava16-7b | 8 | 0.21 ± 0.10 | 2.76 ± 0.19 |
> > > | bkg | mmicl-t5-xxl | 1 | 36.94 ± 0.51 | 12.26 ± 0.68 |
> > > | bkg | mmicl-t5-xxl | 2 | 37.44 ± 0.50 | 34.06 ± 3.37 |
> > > | bkg | mmicl-t5-xxl | 4 | 38.86 ± 0.49 | 38.78 ± 2.81 |
> > > | bkg | mmicl-t5-xxl | 8 | 45.87 ± 1.00 | 39.69 ± 2.09 |
> > > | bkg | qwen-vl-chat | 1 | 20.49 ± 0.37 | 29.80 ± 0.63 |
> > > | bkg | qwen-vl-chat | 2 | 23.80 ± 0.32 | 21.64 ± 0.61 |
> > > | bkg | qwen-vl-chat | 4 | 24.71 ± 0.37 | 23.03 ± 1.14 |
> > > | bkg | qwen-vl-chat | 8 | 28.07 ± 0.66 | 11.69 ± 0.72 |
> > > | bkg | idefics-9b-instruct | 1 | 27.17 ± 0.80 | 7.20 ± 0.73 |
> > > | bkg | idefics-9b-instruct | 2 | 30.20 ± 0.38 | 9.98 ± 0.92 |
> > > | bkg | idefics-9b-instruct | 4 | 34.78 ± 0.13 | 16.09 ± 1.98 |
> > > | bkg | idefics-9b-instruct | 8 | 51.24 ± 1.10 | 19.96 ± 1.82 |
> > > | both | otter-mpt | 1 | 34.44 ± 3.27 | 29.52 ± 3.02 |
> > > | both | otter-mpt | 2 | 13.30 ± 0.29 | 40.99 ± 1.50 |
> > > | both | otter-mpt | 4 | 11.70 ± 0.42 | 44.04 ± 0.63 |
> > > | both | otter-mpt | 8 | 11.20 ± 1.35 | 19.01 ± 2.05 |
> > > | both | llava16-7b | 1 | 96.40 ± 1.92 | 0.27 ± 0.16 |
> > > | both | llava16-7b | 2 | 38.37 ± 7.86 | 4.14 ± 1.73 |
> > > | both | llava16-7b | 4 | 2.05 ± 0.20 | 6.35 ± 0.48 |
> > > | both | llava16-7b | 8 | 1.66 ± 0.05 | 2.10 ± 0.12 |
> > > | both | mmicl-t5-xxl | 1 | 18.66 ± 1.26 | 9.48 ± 0.38 |
> > > | both | mmicl-t5-xxl | 2 | 35.21 ± 0.21 | 34.28 ± 3.72 |
> > > | both | mmicl-t5-xxl | 4 | 33.74 ± 0.18 | 39.30 ± 3.03 |
> > > | both | mmicl-t5-xxl | 8 | 33.69 ± 0.39 | 42.74 ± 1.94 |
> > > | both | qwen-vl-chat | 1 | 42.69 ± 1.45 | 25.60 ± 0.44 |
> > > | both | qwen-vl-chat | 2 | 24.78 ± 0.57 | 23.73 ± 0.81 |
> > > | both | qwen-vl-chat | 4 | 26.11 ± 0.52 | 23.26 ± 1.45 |
> > > | both | qwen-vl-chat | 8 | 23.74 ± 0.55 | 13.09 ± 0.50 |
> > > | both | idefics-9b-instruct | 1 | 47.53 ± 8.39 | 5.24 ± 0.61 |
> > > | both | idefics-9b-instruct | 2 | 20.29 ± 3.68 | 7.61 ± 0.81 |
> > > | both | idefics-9b-instruct | 4 | 11.28 ± 1.43 | 9.62 ± 1.53 |
> > > | both | idefics-9b-instruct | 8 | 8.24 ± 1.07 | 19.63 ± 1.78 |

---

> > > ### Author Response · Authors · 2024-08-28
> > >
> > > # Post 4
> > >
> > > ICL, LOGO, Task: Predict Background
> > >
> > > | Context Contains| Model | Shot | Shape Accuracy (Mean ± SE) | Background Accuracy (Mean ± SE) |
> > > |:----------------|:------|:-----|:-------------------------:|:--------------------------:|
> > > | neither | otter-mpt | 1 | 4.82 ± 0.41 | 35.05 ± 1.65 |
> > > | neither | otter-mpt | 2 | 6.80 ± 0.67 | 51.43 ± 1.34 |
> > > | neither | otter-mpt | 4 | 8.85 ± 0.19 | 51.56 ± 1.83 |
> > > | neither | otter-mpt | 8 | 15.33 ± 0.51 | 28.84 ± 2.36 |
> > > | neither | llava16-7b | 1 | 0.03 ± 0.03 | 1.83 ± 0.79 |
> > > | neither | llava16-7b | 2 | 0.80 ± 0.39 | 2.17 ± 0.65 |
> > > | neither | llava16-7b | 4 | 1.71 ± 0.25 | 5.12 ± 0.08 |
> > > | neither | llava16-7b | 8 | 0.01 ± 0.01 | 0.28 ± 0.03 |
> > > | neither | mmicl-t5-xxl | 1 | 1.10 ± 0.23 | 96.44 ± 0.07 |
> > > | neither | mmicl-t5-xxl | 2 | 1.09 ± 0.18 | 96.63 ± 0.14 |
> > > | neither | mmicl-t5-xxl | 4 | 0.94 ± 0.15 | 96.99 ± 0.06 |
> > > | neither | mmicl-t5-xxl | 8 | 1.56 ± 0.44 | 97.14 ± 0.35 |
> > > | neither | qwen-vl-chat | 1 | 10.26 ± 0.40 | 59.27 ± 1.58 |
> > > | neither | qwen-vl-chat | 2 | 9.69 ± 0.28 | 56.01 ± 0.81 |
> > > | neither | qwen-vl-chat | 4 | 8.84 ± 0.10 | 47.40 ± 0.28 |
> > > | neither | qwen-vl-chat | 8 | 8.76 ± 0.32 | 57.80 ± 0.76 |
> > > | neither | idefics-9b-instruct | 1 | 8.27 ± 1.52 | 72.12 ± 2.24 |
> > > | neither | idefics-9b-instruct | 2 | 4.25 ± 1.15 | 69.48 ± 1.04 |
> > > | neither | idefics-9b-instruct | 4 | 6.28 ± 0.74 | 74.93 ± 1.43 |
> > > | neither | idefics-9b-instruct | 8 | 12.53 ± 0.91 | 78.97 ± 0.52 |
> > > | shape | otter-mpt | 1 | 8.35 ± 0.40 | 41.75 ± 1.61 |
> > > | shape | otter-mpt | 2 | 10.54 ± 0.22 | 57.60 ± 2.14 |
> > > | shape | otter-mpt | 4 | 11.07 ± 0.34 | 55.03 ± 2.24 |
> > > | shape | otter-mpt | 8 | 15.43 ± 0.84 | 29.15 ± 2.08 |
> > > | shape | llava16-7b | 1 | 0.41 ± 0.20 | 1.94 ± 0.21 |
> > > | shape | llava16-7b | 2 | 1.60 ± 0.77 | 2.47 ± 0.75 |
> > > | shape | llava16-7b | 4 | 2.56 ± 0.13 | 4.17 ± 0.38 |
> > > | shape | llava16-7b | 8 | 0.15 ± 0.05 | 0.59 ± 0.10 |
> > > | shape | mmicl-t5-xxl | 1 | 0.87 ± 0.20 | 95.09 ± 0.15 |
> > > | shape | mmicl-t5-xxl | 2 | 0.78 ± 0.11 | 96.84 ± 0.04 |
> > > | shape | mmicl-t5-xxl | 4 | 0.86 ± 0.18 | 97.12 ± 0.10 |
> > > | shape | mmicl-t5-xxl | 8 | 1.11 ± 0.30 | 97.16 ± 0.30 |
> > > | shape | qwen-vl-chat | 1 | 10.10 ± 0.36 | 62.68 ± 0.92 |
> > > | shape | qwen-vl-chat | 2 | 9.44 ± 0.30 | 60.66 ± 0.41 |
> > > | shape | qwen-vl-chat | 4 | 9.66 ± 0.29 | 50.37 ± 0.17 |
> > > | shape | qwen-vl-chat | 8 | 8.29 ± 0.43 | 58.95 ± 0.60 |
> > > | shape | idefics-9b-instruct | 1 | 11.19 ± 0.93 | 65.80 ± 2.86 |
> > > | shape | idefics-9b-instruct | 2 | 8.71 ± 0.85 | 73.92 ± 1.42 |
> > > | shape | idefics-9b-instruct | 4 | 9.07 ± 0.74 | 79.31 ± 0.82 |
> > > | shape | idefics-9b-instruct | 8 | 12.67 ± 0.61 | 80.37 ± 0.47 |
> > > | bkg | otter-mpt | 1 | 6.53 ± 0.52 | 50.84 ± 4.49 |
> > > | bkg | otter-mpt | 2 | 9.24 ± 0.70 | 34.89 ± 0.69 |
> > > | bkg | otter-mpt | 4 | 10.46 ± 0.29 | 29.80 ± 1.84 |
> > > | bkg | otter-mpt | 8 | 14.76 ± 0.74 | 21.05 ± 0.99 |
> > > | bkg | llava16-7b | 1 | 0.09 ± 0.05 | 96.54 ± 1.76 |
> > > | bkg | llava16-7b | 2 | 0.61 ± 0.31 | 59.13 ± 5.51 |
> > > | bkg | llava16-7b | 4 | 2.06 ± 0.13 | 23.68 ± 1.36 |
> > > | bkg | llava16-7b | 8 | 0.03 ± 0.01 | 4.13 ± 0.12 |
> > > | bkg | mmicl-t5-xxl | 1 | 1.84 ± 0.45 | 51.10 ± 2.79 |
> > > | bkg | mmicl-t5-xxl | 2 | 2.04 ± 0.34 | 92.63 ± 0.44 |
> > > | bkg | mmicl-t5-xxl | 4 | 2.27 ± 0.39 | 92.88 ± 0.42 |
> > > | bkg | mmicl-t5-xxl | 8 | 4.12 ± 1.05 | 91.33 ± 0.67 |
> > > | bkg | qwen-vl-chat | 1 | 6.68 ± 0.50 | 90.22 ± 0.98 |
> > > | bkg | qwen-vl-chat | 2 | 5.28 ± 0.23 | 88.98 ± 0.48 |
> > > | bkg | qwen-vl-chat | 4 | 5.56 ± 0.08 | 84.31 ± 0.16 |
> > > | bkg | qwen-vl-chat | 8 | 6.15 ± 0.25 | 83.11 ± 0.27 |
> > > | bkg | idefics-9b-instruct | 1 | 12.71 ± 0.91 | 68.46 ± 2.43 |
> > > | bkg | idefics-9b-instruct | 2 | 7.33 ± 1.22 | 63.89 ± 2.18 |
> > > | bkg | idefics-9b-instruct | 4 | 8.81 ± 0.98 | 53.59 ± 1.42 |
> > > | bkg | idefics-9b-instruct | 8 | 16.28 ± 0.75 | 56.62 ± 2.00 |
> > > | both | otter-mpt | 1 | 6.06 ± 0.57 | 51.97 ± 4.45 |
> > > | both | otter-mpt | 2 | 12.21 ± 0.17 | 33.02 ± 1.77 |
> > > | both | otter-mpt | 4 | 12.65 ± 0.36 | 25.80 ± 1.94 |
> > > | both | otter-mpt | 8 | 14.53 ± 0.79 | 21.70 ± 0.98 |
> > > | both | llava16-7b | 1 | 0.17 ± 0.08 | 96.29 ± 1.88 |
> > > | both | llava16-7b | 2 | 1.16 ± 0.57 | 57.44 ± 5.89 |
> > > | both | llava16-7b | 4 | 2.13 ± 0.08 | 16.99 ± 1.12 |
> > > | both | llava16-7b | 8 | 0.46 ± 0.06 | 4.08 ± 0.48 |
> > > | both | mmicl-t5-xxl | 1 | 0.90 ± 0.22 | 35.86 ± 2.37 |
> > > | both | mmicl-t5-xxl | 2 | 1.44 ± 0.31 | 92.37 ± 0.62 |
> > > | both | mmicl-t5-xxl | 4 | 1.71 ± 0.38 | 92.05 ± 0.52 |
> > > | both | mmicl-t5-xxl | 8 | 2.54 ± 0.56 | 91.65 ± 0.16 |
> > > | both | qwen-vl-chat | 1 | 4.27 ± 0.38 | 93.79 ± 0.60 |
> > > | both | qwen-vl-chat | 2 | 5.73 ± 0.20 | 89.17 ± 0.34 |
> > > | both | qwen-vl-chat | 4 | 6.65 ± 0.05 | 85.28 ± 0.34 |
> > > | both | qwen-vl-chat | 8 | 5.80 ± 0.27 | 82.84 ± 0.68 |
> > > | both | idefics-9b-instruct | 1 | 9.66 ± 0.80 | 74.84 ± 2.30 |
> > > | both | idefics-9b-instruct | 2 | 10.12 ± 0.84 | 64.73 ± 3.10 |
> > > | both | idefics-9b-instruct | 4 | 11.80 ± 0.78 | 61.29 ± 1.47 |
> > > | both | idefics-9b-instruct | 8 | 15.85 ± 0.76 | 56.28 ± 2.01 |

---

> > > ### Author Response · Authors · 2024-08-28
> > >
> > > # Post 5
> > >
> > > ICL, LOGO, Task: Predict Both
> > > | Context Contains | Model | Shot | Shape Accuracy (Mean ± SE) | Background Accuracy (Mean ± SE) |
> > > |:----------------|:------|:-----|:-------------------------:|:--------------------------:|
> > > | neither | otter-mpt | 1 | 20.50 ± 1.00 | 60.25 ± 1.99 |
> > > | neither | otter-mpt | 2 | 23.53 ± 0.45 | 76.27 ± 2.09 |
> > > | neither | otter-mpt | 4 | 22.44 ± 0.59 | 75.36 ± 2.21 |
> > > | neither | otter-mpt | 8 | 20.37 ± 1.18 | 37.33 ± 2.15 |
> > > | neither | llava16-7b | 1 | 3.33 ± 0.46 | 4.80 ± 1.18 |
> > > | neither | llava16-7b | 2 | 1.93 ± 0.33 | 2.26 ± 0.88 |
> > > | neither | llava16-7b | 4 | 1.31 ± 0.13 | 4.19 ± 0.59 |
> > > | neither | llava16-7b | 8 | 0.21 ± 0.03 | 0.25 ± 0.05 |
> > > | neither | mmicl-t5-xxl | 1 | 30.43 ± 2.34 | 97.65 ± 0.38 |
> > > | neither | mmicl-t5-xxl | 2 | 32.48 ± 2.04 | 97.94 ± 0.09 |
> > > | neither | mmicl-t5-xxl | 4 | 33.74 ± 1.67 | 97.99 ± 0.10 |
> > > | neither | mmicl-t5-xxl | 8 | 37.31 ± 2.00 | 98.34 ± 0.07 |
> > > | neither | qwen-vl-chat | 1 | 20.22 ± 0.49 | 67.09 ± 0.55 |
> > > | neither | qwen-vl-chat | 2 | 21.49 ± 0.40 | 69.64 ± 0.70 |
> > > | neither | qwen-vl-chat | 4 | 23.43 ± 0.28 | 63.94 ± 0.89 |
> > > | neither | qwen-vl-chat | 8 | 25.73 ± 0.51 | 73.49 ± 1.67 |
> > > | neither | idefics-9b-instruct | 1 | 28.39 ± 1.13 | 66.08 ± 1.81 |
> > > | neither | idefics-9b-instruct | 2 | 29.87 ± 0.93 | 73.97 ± 1.67 |
> > > | neither | idefics-9b-instruct | 4 | 33.80 ± 0.84 | 85.87 ± 1.29 |
> > > | neither | idefics-9b-instruct | 8 | 48.66 ± 1.51 | 90.67 ± 0.80 |
> > > | shape | otter-mpt | 1 | 24.19 ± 0.83 | 66.87 ± 1.86 |
> > > | shape | otter-mpt | 2 | 15.90 ± 0.62 | 80.17 ± 1.45 |
> > > | shape | otter-mpt | 4 | 11.70 ± 0.68 | 80.93 ± 1.75 |
> > > | shape | otter-mpt | 8 | 13.25 ± 0.59 | 37.90 ± 1.95 |
> > > | shape | llava16-7b | 1 | 74.34 ± 11.86 | 4.10 ± 1.17 |
> > > | shape | llava16-7b | 2 | 32.56 ± 7.17 | 2.14 ± 0.70 |
> > > | shape | llava16-7b | 4 | 3.10 ± 0.19 | 4.19 ± 0.46 |
> > > | shape | llava16-7b | 8 | 1.33 ± 0.11 | 0.31 ± 0.05 |
> > > | shape | mmicl-t5-xxl | 1 | 27.85 ± 2.43 | 97.46 ± 0.46 |
> > > | shape | mmicl-t5-xxl | 2 | 28.92 ± 1.90 | 97.79 ± 0.14 |
> > > | shape | mmicl-t5-xxl | 4 | 29.77 ± 1.63 | 97.96 ± 0.09 |
> > > | shape | mmicl-t5-xxl | 8 | 30.64 ± 1.27 | 98.13 ± 0.08 |
> > > | shape | qwen-vl-chat | 1 | 28.95 ± 0.54 | 68.91 ± 0.94 |
> > > | shape | qwen-vl-chat | 2 | 25.04 ± 0.33 | 70.83 ± 0.32 |
> > > | shape | qwen-vl-chat | 4 | 25.57 ± 0.38 | 65.44 ± 0.86 |
> > > | shape | qwen-vl-chat | 8 | 26.15 ± 0.28 | 71.08 ± 1.39 |
> > > | shape | idefics-9b-instruct | 1 | 34.39 ± 3.44 | 56.65 ± 4.48 |
> > > | shape | idefics-9b-instruct | 2 | 21.30 ± 3.01 | 69.06 ± 2.64 |
> > > | shape | idefics-9b-instruct | 4 | 11.96 ± 1.51 | 85.00 ± 1.11 |
> > > | shape | idefics-9b-instruct | 8 | 8.16 ± 0.61 | 91.63 ± 0.35 |
> > > | bkg | otter-mpt | 1 | 21.95 ± 0.67 | 66.14 ± 0.63 |
> > > | bkg | otter-mpt | 2 | 29.38 ± 0.32 | 54.60 ± 1.53 |
> > > | bkg | otter-mpt | 4 | 27.42 ± 0.52 | 41.08 ± 3.58 |
> > > | bkg | otter-mpt | 8 | 22.60 ± 0.92 | 26.06 ± 2.21 |
> > > | bkg | llava16-7b | 1 | 4.94 ± 0.93 | 69.59 ± 15.21 |
> > > | bkg | llava16-7b | 2 | 2.26 ± 0.50 | 33.93 ± 3.65 |
> > > | bkg | llava16-7b | 4 | 1.70 ± 0.11 | 18.66 ± 2.03 |
> > > | bkg | llava16-7b | 8 | 0.20 ± 0.04 | 3.25 ± 0.18 |
> > > | bkg | mmicl-t5-xxl | 1 | 34.65 ± 0.92 | 91.11 ± 1.68 |
> > > | bkg | mmicl-t5-xxl | 2 | 36.41 ± 0.75 | 95.24 ± 0.30 |
> > > | bkg | mmicl-t5-xxl | 4 | 39.11 ± 0.85 | 95.92 ± 0.16 |
> > > | bkg | mmicl-t5-xxl | 8 | 44.62 ± 1.91 | 95.87 ± 0.17 |
> > > | bkg | qwen-vl-chat | 1 | 20.14 ± 0.56 | 93.70 ± 0.42 |
> > > | bkg | qwen-vl-chat | 2 | 23.73 ± 0.47 | 89.59 ± 0.25 |
> > > | bkg | qwen-vl-chat | 4 | 24.06 ± 0.31 | 84.28 ± 0.46 |
> > > | bkg | qwen-vl-chat | 8 | 26.60 ± 0.59 | 83.47 ± 0.36 |
> > > | bkg | idefics-9b-instruct | 1 | 26.17 ± 1.00 | 83.34 ± 1.50 |
> > > | bkg | idefics-9b-instruct | 2 | 30.57 ± 0.65 | 63.00 ± 3.17 |
> > > | bkg | idefics-9b-instruct | 4 | 36.51 ± 0.63 | 52.21 ± 2.36 |
> > > | bkg | idefics-9b-instruct | 8 | 55.42 ± 0.99 | 44.65 ± 0.77 |
> > > | both | otter-mpt | 1 | 34.53 ± 1.32 | 62.63 ± 0.74 |
> > > | both | otter-mpt | 2 | 21.25 ± 0.58 | 47.56 ± 1.66 |
> > > | both | otter-mpt | 4 | 14.52 ± 0.62 | 37.55 ± 3.02 |
> > > | both | otter-mpt | 8 | 14.68 ± 0.83 | 24.46 ± 1.82 |
> > > | both | llava16-7b | 1 | 80.99 ± 10.22 | 79.58 ± 11.81 |
> > > | both | llava16-7b | 2 | 33.19 ± 7.18 | 38.99 ± 4.22 |
> > > | both | llava16-7b | 4 | 2.95 ± 0.32 | 14.74 ± 1.25 |
> > > | both | llava16-7b | 8 | 1.43 ± 0.09 | 2.97 ± 0.12 |
> > > | both | mmicl-t5-xxl | 1 | 35.45 ± 1.90 | 82.40 ± 3.61 |
> > > | both | mmicl-t5-xxl | 2 | 33.89 ± 0.64 | 95.02 ± 0.28 |
> > > | both | mmicl-t5-xxl | 4 | 33.67 ± 0.30 | 95.01 ± 0.19 |
> > > | both | mmicl-t5-xxl | 8 | 34.30 ± 0.39 | 94.82 ± 0.16 |
> > > | both | qwen-vl-chat | 1 | 60.70 ± 1.68 | 94.22 ± 0.37 |
> > > | both | qwen-vl-chat | 2 | 27.58 ± 0.37 | 88.56 ± 0.26 |
> > > | both | qwen-vl-chat | 4 | 27.50 ± 0.27 | 84.86 ± 0.26 |
> > > | both | qwen-vl-chat | 8 | 27.08 ± 0.11 | 81.34 ± 0.19 |
> > > | both | idefics-9b-instruct | 1 | 58.31 ± 7.93 | 90.35 ± 1.99 |
> > > | both | idefics-9b-instruct | 2 | 23.97 ± 3.22 | 67.90 ± 2.86 |
> > > | both | idefics-9b-instruct | 4 | 13.44 ± 1.57 | 55.75 ± 3.37 |
> > > | both | idefics-9b-instruct | 8 | 10.24 ± 0.74 | 42.01 ± 0.79 |

---

> > ### Author Response · Authors · 2024-08-28
> >
> > # Post 6
> >
> > Zero-Shot, LOGO
> > | Model | Shape Accuracy (Mean ± SE) | Background Accuracy (Mean ± SE) |
> > |:---|:-------------------------:|:--------------------------:|
> > | Blipv2-t5 | 23.92 ± 0.52 | 48.62 ± 0.83 |
> > | CogVLM | 21.18 ± 0.74 | 66.28 ± 1.11 |
> > | InstructBlip-7B | 24.35 ± 1.31 | 7.5 ± 0.74 |
> > | llava1.5-7B | 23.77 ± 0.86 | 54.16 ± 0.93 |
> > | QWen | 20.82 ± 0.92 | 33.72 ± 1.15|

---

> > > ### Comment · Reviewer_cHDM · 2024-08-31
> > >
> > > I appreciate the comprehensive rebuttal which fully addresses my concerns. I'd like to change my rating to **7: Good paper, accept** - however, OpenReview doesn't let me edit my initial review (only the successive comments). I will additionally ping the AC via the committee chat.

---

### Official Review · Reviewer_UDvg · 2024-07-24

**Rating:** 7
**Confidence:** 3
**Correctness:** Yes
**Clarity:** No. See weaknesses.

**Review:**

The paper makes a valuable contribution by introducing a novel benchmark for evaluating the abstract shape recognition capabilities of vision models. The IllusionBench dataset fills a gap in existing benchmarks and can potentially drive future research in improving the robustness of visual perception systems. However, the paper would benefit from addressing the identified weaknesses, particularly in terms of providing a clearer overview, simplifying notations, and resolving formatting issues. With these improvements, the paper has the potential to be a strong addition to the NeurIPS proceedings.

**Strengths:**

- The paper presents a comprehensive benchmark for evaluating the abstract shape recognition skills of computer vision models. The dataset's focus on abstract shapes makes it applicable to a wide range of vision models beyond multimodal VLMs.
- The authors have made the IllusionBench dataset publicly available on Hugging Face, facilitating its use by the research community.
- The experiments conducted in the paper provide valuable insights into the limitations of current multimodal VLMs in recognizing abstract shapes, highlighting important avenues for future research.

**Additional Feedback:**

N/A

**Documentation:**

Yes

**Limitations:**

Yes

**Opportunities For Improvement:**

The paper seems to be completed in a rush and the writing needs to be improved.

- Section 3 would benefit from a high-level overview before delving into the technical details and notations, to help readers grasp the overall structure and purpose of the datasets.
- The notation introduced in Section 3.1 appears unnecessarily complicated, which may hinder the reader's understanding of the dataset generation process.
- There are some formatting and typographical issues in the paper, such as the reference to a missing section ("see ??") on line 124 and the low resolution of Figure 8 in the appendix.
- The appendix on page 20 has formatting problems that need to be addressed

**Relation To Prior Work:**

Yes

**Summary And Contributions:**

This paper introduces IllusionBench, a collection of three datasets designed to evaluate the ability of vision-language models (VLMs) to recognize abstract shapes represented by arrangements of elements in complex visual scenes. The authors conduct experiments to assess the shape recognition capabilities of state-of-the-art VLMs in zero-shot, few-shot, and domain generalization settings. The results reveal significant limitations in the shape perception abilities of current VLMs, with models often focusing more on scene elements rather than the intended abstract shapes.

---

> ### Author Rebuttal · Authors · 2024-08-17
>
> We thank the reviewer for their positive assessment of our work. In particular, for praising how our dataset **fills a gap in the field of abstract shape recognition, and how it could drive the research and development of better models**.
>
> We agree with the reviewer that the presentation of our work could be further improved, for these reasons we have updated our draft (not visible in openreview) as follows:
> - **Better overview**: we introduced a brief paragraph at the beginning of Section 3, that helps the reader navigate the section better and understand the high level motivation behind each dataset.
> - **Notation too complex**: we agree, and we have simplified the notation as follows. Although simple, these changes significantly improve the readability:
>    1. $x_i^C \to x_i $
>    2. $y_i^C \to c_i$
>    3. $y_j^S \to s_j$
>    4. $p_T \to p_S$ (for consistency)
> - **Unclear dataset generation**: In order to further clarify how the dataset was generated, we will introduce a diagram (shown in the pdf we attached to the rebuttal) that will help the readers to immediately understand the generative process.
>
> We have also fixed all typographical and formatting issues indicated.
>
> We hope the reviewer will agree these modifications contribute to a better presentation of our work, and we are happy to receive suggestions to improve it further.

---

> > ### Comment · Reviewer_UDvg · 2024-08-29
> >
> > Thanks for the response. I appreciate these efforts and decide to keep my rating.

---

### Decision · Program_Chairs · 2024-09-26

**Decision:**

Accept (Poster)

**Comment:**

This paper initially received mixed review scores: 7, 5, 7. Reviewers overall well recognized the value of this work, regarding it a novel benchmark for evaluating the abstract shape recognition capabilities of vision models and potentially driving future research in improving the robustness of visual perception systems (as summarized by UDvg). And the paper is also well written and easy to follow. The reviewers also added some comments for improvement (mainly from cHDM), such as further writing improvement, human evaluation and priming, prompt sensitivity, response processing clarification, codebase refactorization, dataset naming, etc. The rebuttal was persuasive and addressed most concerns. After rebuttal, the second Reviewer UDvg increased the score to 7 (noted in his/her comment). The final ratings unanimously recommend acceptance. The AC checked the paper, rebuttal, and review comments, and recommends accepting the paper.